# Deciphering the molecular mechanisms of mother-to-egg immune protection in the mealworm beetle *Tenebrio molitor*

**Guillaume Tetreau**[1¤]*, **Julien Dhinaut**[2], **Richard Galinier**[1], **Pascaline Audant-Lacour**[3‡], **Sébastien N. Voisin**[4], **Karim Arafah**[4], **Manon Chogne**[2], **Frédérique Hilliou**[3], **Anaïs Bordes**[1], **Camille Sabarly**[2], **Philippe Chan**[5], **Marie-Laure Walet-Balieu**[5], **David Vaudry**[5], **David Duval**[1], **Philippe Bulet**[4,6], **Christine Coustau**[3], **Yannick Moret**[2], **Benjamin Gourbal**[1]*

**1** IHPE, Univ. Montpellier, CNRS, Ifremer, Univ. Perpignan Via Domitia, Perpignan, France, **2** Équipe Écologie Évolutive, UMR CNRS 6282 BioGéoSciences, Université Bourgogne-Franche Comté, Dijon, France, **3** CNRS, INRAE, Université Nice Côte d'Azur, UMR 1355–7254 Institut Sophia Agrobiotech, Sophia Antipolis, France, **4** Plateforme BioPark d'Archamps, ArchParc, Saint Julien en Genevois, France, **5** PISSARO Proteomic Platform, Institute for Research and Innovation in Biomedicine, University of Rouen, Rouen, France, **6** CR Université Grenoble Alpes, Institute for Advanced Biosciences, INSERM U1209, CNRS UMR5309, La Tronche, France

¤ Current address: Univ. Grenoble Alpes, CNRS, CEA, IBS, F-38000 Grenoble, France
‡ Authorship confirmed by corresponding author
* guillaume.tetreau@gmail.com (GT); benjamin.gourbal@univ-perp.fr (BG)

**Data Availability Statement:** Our reference transcriptome, its proteome and annotation were used as a resource for candidate genes and proteins in the following experiments. They are

## Abstract

In a number of species, individuals exposed to pathogens can mount an immune response and transmit this immunological experience to their offspring, thereby protecting them against persistent threats. Such vertical transfer of immunity, named trans-generational immune priming (TGIP), has been described in both vertebrates and invertebrates. Although increasingly studied during the last decade, the mechanisms underlying TGIP in invertebrates are still elusive, especially those protecting the earliest offspring life stage, *i.e.* the embryo developing in the egg. In the present study, we combined different proteomic and transcriptomic approaches to determine whether mothers transfer a "signal" (such as fragments of infecting bacteria), mRNA and/or protein/peptide effectors to protect their eggs against two natural bacterial pathogens, namely the Gram-positive *Bacillus thuringiensis* and the Gram-negative *Serratia entomophila*. By taking the mealworm beetle *Tenebrio molitor* as a biological model, our results suggest that eggs are mainly protected by an active direct transfer of a restricted number of immune proteins and of antimicrobial peptides. In contrast, the present data do not support the involvement of mRNA transfer while the transmission of a "signal", if it happens, is marginal and only occurs within 24h after maternal exposure to bacteria. This work exemplifies how combining global approaches helps to disentangle the different scenarios of a complex trait, providing a comprehensive characterization of TGIP mechanisms in *T. molitor*. It also paves the way for future alike studies focusing on TGIP in a wide range of invertebrates and vertebrates to identify additional candidates that could be specific to TGIP and to investigate whether the TGIP mechanisms found herein are specific or common to all insect species.

available for download on the IHPE laboratory website (http://ihpe.univ-perp.fr/acces-aux-donnees). The data are also available on the NCBI database under the BioProject ID PRJNA646689 with SRA numbers SRR12235350 and SRR12235349. The mass spectrometry proteomics data have been deposited to the ProteomeXchange Consortium via the PRIDE partner repository with the dataset identifier PXD018772. All other relevant data are within the manuscript and its Supporting Information files.

**Funding:** This work was funded by the MATER-IMMUNITY Project (ANR-14-CE02-0009) from the French National Research Agency (ANR; https://anr.fr/) granted to YM, BG and CC. The PISSARO Proteomic Platform gets funding from Normandy Region (https://www.normandie.fr/) and ERDF (https://www.enedis.fr/). Peptidomics and a part of proteomics studies were funded by the R&D budget from the Plateforme BioPark d'Archamps (http://www.biopark-archamps.org/). The funders had no role in study design, data collection and analysis, decision to publish, or preparation of the manuscript.

**Competing interests:** The authors declare that they have no competing interest. Pascaline Audant-Lacour was unable to confirm her authorship contributions. On her behalf, all other authors have reported her contributions to the best of their knowledge.

## Author summary

All living organisms are regularly exposed to a wide and diverse range of pathogens. To protect themselves, many species have developed an immune system able to detect and eradicate these pathogens. Most interestingly, this immunological experience can be transferred by parents to their offspring to protect them from pathogens that may persist in the environment and to which they could be exposed during their life. While extensively studied in vertebrates, this phenomenon–called trans-generational immune priming (TGIP)–has only been identified a decade ago in invertebrates and the supporting molecular mechanisms are still largely unknown. Recently, we proposed four different scenarios as a practical framework to investigate the mechanisms supporting this complex phenomenon. In the present study, we combined different molecular approaches to disentangle these different scenarios and provide a comprehensive characterization of maternal TGIP mechanisms in a model insect, the mealworm beetle *Tenebrio molitor*.

## Introduction

Within the life span of an individual, its past experience of infection has an impact on its capacity to respond better to a new encounter with the same pathogen. In invertebrates, several recent studies have provided evidence that the innate immune system could be "primed" in a sustainable manner, increasing the protection against secondary infections with a pathogen already encountered [1–4]. Interestingly, such immunological experience can also be transferred from the parent(s) to the offspring, resulting in a vertical transfer of immunity named trans-generational immune priming (TGIP) [2,5–7].

TGIP studies essentially focused on arthropods, and within arthropods most of the work has been conducted on insects [6]. Multiple studies revealed an enhanced immune activity and/or an increased survival to infection in juveniles [5,8–11] and adult offspring [12–15] when parents had been exposed to pathogens (bacteria, viruses, fungi, metazoan parasites) [6]. Indeed, TGIP provides progeny with protection against pathogens that persist across generations before their own immune system becomes mature. Interestingly, such transferred protection can occur very early in offspring life, and notably in eggs [16–18]. The defense of eggs against pathogens is the primary and fundamental step in the acquisition of TGIP, which allows protecting embryos before they can protect themselves.

Egg protection in insects has received an increasing attention. It has been suggested to occur through endogenous egg defenses by different mechanisms, triggering differential expression of immune-related genes or through parental transfer of protection by the direct transmission of immune effectors to the eggs [6,13,19–22]. While maternal transgenerational protection has been extensively studied [6], evidence of paternal offspring immune protection has also been provided [23,24], although the underlying mechanisms might differ. Four scenarios have been proposed to formalize the different forms that such transmission of parental immunological experience may take [6]. Parents could transfer a "signal" stimulating the egg immune system (scenario 1), which could for example be peptides from infecting bacteria directly stored by mothers in the eggs to activate embryo endogenous immune response [25]. Alternatively, parents could transfer mRNA (scenario 2) or directly the immune protein/peptide effectors (scenario 3) that confer a passive protection to the eggs. Lastly, epigenetic modifications (scenario 4) might trigger an increased expression of immune candidate genes in order to deal with persistent pathogens in the environment [26].

The involvement of each (or both) of these processes has not been fully elucidated from a functional point of view yet. In insects, some known molecular effectors of the immunological response such as lysozyme, antimicrobial peptides (AMPs) or phenoloxidase (PO) have been identified and their expression quantified using reverse transcription-quantitative PCR (RT-qPCR) in the context of TGIP (see [6] for review). However, this supposes that trans-generational immune priming relies on the same mechanisms as the within-individual immune priming. Only a limited number of studies used global approaches to unravel the potential role of other genes and proteins in TGIP, by using RNA-seq approach [27,28], or by proteomics profiling (SDS-PAGE [8,16] or 2D-PAGE [29]). Despite these studies, TGIP still raises considerable questions related to the molecular mechanisms and transfer processes, especially those resulting in egg protection.

The mealworm beetle, *Tenebrio molitor*, is probably the insect for which TGIP has been best described so far [30]. TGIP effects were revealed through enhanced immune activity in primed eggs [16,17,19,31], larvae [5] and adult offspring [13,15]. Enhanced immunity in the offspring may result from the immune priming of either fathers or mothers, although each parental effect involves the enhancement of different immune effectors in the offspring at the adult stage [13]. TGIP of the offspring does not appear to be pathogen-specific as the offspring of mothers primed with the Gram-negative bacteria, *Serratia entomophila*, and those primed with the Gram-positive bacteria, *Bacillus thuringiensis*, had a similar enhanced survival to bacterial infection, although different immune components appear to be affected [15]. Bacterially immune primed females provide enhanced antibacterial activity to eggs that are produced from the second to the eighth day after the maternal priming only [17]. While egg protection may rely on the transfer of maternal immune effectors to the egg or/and the induction of egg immune genes, it appears that enhanced egg immunity following a maternal immune priming in *T. molitor* is achieved by both of these mechanisms but in a pathogen-dependent manner [19]. Indeed, depending on the pathogen used for the maternal immune priming, levels of antibacterial activity in primed eggs could be strong soon after being laid, and then decreasing until hatching occurs which is consistent with the hypothesis that mothers could passively supply their eggs with antibacterial substances, which are then metabolized as eggs develop. By contrast, primed eggs could have very low levels of antibacterial activity soon after being laid and then exhibit increased levels to reach a plateau from the third day post-oviposition to hatching. This latter pattern of variation of antibacterial activity with egg age would suggest that antibacterial substances are synthesized in the eggs [32] and not directly provided by the mother. However, while these antibacterial substances, which are likely proteins [16], could be produced by the eggs, it is not known whether the transcripts at the origin of these antibacterial proteins are synthesized in the eggs too or produced and transferred by mothers to the eggs. There is therefore a crucial need to use global approaches to unravel the functional mechanisms of egg immune protection.

Here, we studied the TGIP response in eggs of *T. molitor* towards different maternal immune stimulations by two natural bacterial pathogens, namely the Gram-positive *B. thuringiensis* (*Bt*) and the Gram-negative *S. entomophila* (*Se*) [30], using integrative omics approaches and functional validation. We revealed that a complex balance between parental input and egg endogenous defenses is needed to support efficient egg defense mechanisms. Notably, our results support that egg protection essentially relies on the direct transfer of protein and peptide effectors (scenario 3), irrespective of the bacteria used for mother priming, while the transfer of mRNA is excluded (scenario 2). "Signal(s)" might be transferred (scenario 1) but their nature is unknown yet and their effect is marginal and only occurs within 24h after maternal exposure to bacteria. Epigenetic marks (scenario 4) could not be studied, as it should be investigated by dedicated approaches in future studies. The different scenarios of such a

complex trait could be disentangled by a combination of global comparative approaches and it paves the way for future alike studies focusing on TGIP in a wide range of invertebrates and vertebrates to identify additional candidates that could be specific to TGIP.

## Results & discussion

### Experimental approaches to decipher TGIP mechanisms

Maternal TGIP was studied in the mealworm beetle *T. molitor* through four complementary experiments (Fig 1). These experiments involve: (1) the production of a reference transcriptome database enriched with defense-related transcripts to allow identification of candidates from the proteomic approach (in the absence of a genome of *T. molitor*), (2) the identification of the main proteins and peptides differentially abundant in ovaries and eggs of primed mothers through global proteomic and peptidomic approaches, (3) the functional invalidation of candidate immune effectors involved in eggs immune protection through RNA interference (RNAi) and (4) the determination of the temporal dynamics of the production of candidate immune effector transcripts in primed mothers and eggs. While experiment 1 provides a high-quality resource for the proper identification of candidate genes, experiments 2 and 3 allow

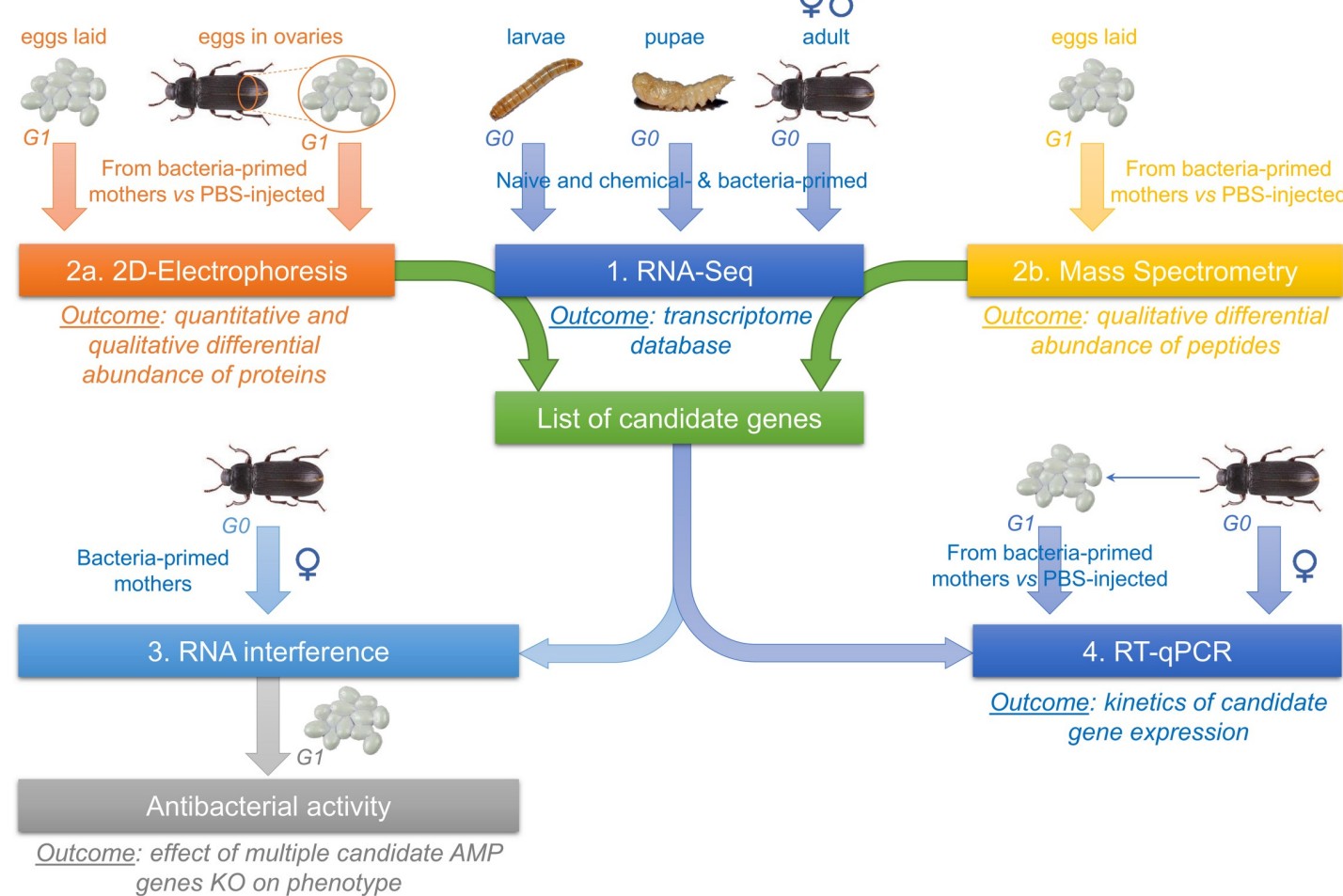

**Fig 1. Summary of the different experimental approaches used to characterize TGIP mechanisms in *T. molitor*.**

identifying the effector immune proteins directly transferred by mothers to the eggs and validate their involvement in the observed phenotype, respectively. Finally, kinetics of candidate gene expression (experiment 4) allows determining if mRNAs are also transferred and/or if eggs express their own set of effectors, depending on their age and on the time elapsed since mother priming. Altogether, these experiments aim at deciphering which scenario(s) of TGIP is at play in *T. molitor* [6].

## The *de novo* transcriptome of *T. molitor* is a high-quality resource for the identification of candidate genes and proteins

A transcriptome dataset was generated by Illumina sequencing of a total of 150 individuals of *T. molitor* at different developmental stages (egg, larval instars, adults), sex (male, female) and physiological conditions (chemically- or bacteriologically-primed or naive). Our *de novo* transcriptome assembly resulted in 110,963 transcripts (Table 1). The transcript lengths ranged from 201 to 26,158bp and the ExN50 length was 1,135 bp. Our transcriptome was searched for conserved eukaryotic gene sets and 94.76% of CEGMA genes were identified as full length, confirming the completeness of our dataset. The use of FrameDP resulted in 45,505 predicted proteins (Table 1), among which 50.93% were full-length proteins. Of these predicted proteins of *T. molitor*, 80% were successfully aligned to *Tribolium castaneum* proteome with a minimum e-value of $10^{-5}$ and annotated (Table 1). This protein annotation percentage of our transcriptome is higher than a previous *T. molitor* transcriptome (56.29%, [33]) and than other RNAseq data from coleopteran where proteins annotation ranged from 64.4% to 73.0% of the total number of proteins detected [34–36]. The longest transcript (26 kb) of our transcriptome assembly was not the result of a chimeric assembly but was identified as a projectin [37]. The presence of transcripts involved in immunity was specifically searched based on functional annotations obtained from InterProScan. The automatically annotated transcripts displayed

**Table 1. Statistics of our transcriptome of *T. molitor* compared to those from other coleopteran reported in previous studies.**

|  | Tenebrio molitor | Tenebrio molitor | Tribolium castaneum | Microdera punctipennis | Batocera horsfieldi |
|---|---|---|---|---|---|
| **General information** |  |  |  |  |  |
| Order:Family | Coleoptera: Tenebrionidae | Coleoptera: Tenebrionidae | Coleoptera: Tenebrionidae | Coleoptera: Tenebrionidae | Coleoptera: Cerambycidae |
| Source | Present study | [33] | [36] | [34] | [35] |
| **Reads** |  |  |  |  |  |
| Raw reads | 243,167,388 | 95,339,034 | 7,315,113 | 48,158,004 | 51,908,500 |
| Clean reads | 203,673,036 | / | 4,610,640 | 39,654,340 | 50,028,651 |
| **Transcripts** |  |  |  |  |  |
| Total number | 110,963 | 90,956 | 13,243 | 56,635 | 171,664 |
| < 1000 bp | 87,461 (78.8%) | / | 5,558 (42.0%) | 44,780 (79,1%) | 123,295 (71.8%) |
| < 2000 bp | 102,412 (92.3%) | / | / | / | 141,883 (82.7%) |
| < 3000 bp | 107,825 (97.2%) | / | 11,928 (90.0%) | 54,621 (96,4%) | / |
| Minimum length | 201 | / | 81 | 89 | 201 |
| Maximum length | 26,158 | / | 63,354 | 10,230 | 27,920 |
| Mean length | 741 | / | / | 666 | 1,188 |
| Median length | 400 | / | / | / | / |
| N50 (ExN50) | 1,261 (1,135) | 1,644 | / | 1,603 | 3,143 |
| % GC | 41.34 | / | / | / | 42.88 |
| **Proteins** |  |  |  |  |  |
| Predicted Proteins | 45,505 | 77,118 | / | 56,344 | 87,743 |
| Annotated proteins | 36,412 (80.0%) | 51,130 (66.3%) | / | 41,109 (73.0%) | 56,507 (64.4%) |

proteases and protease inhibitors, pattern recognition molecules, elements of TOLL pathway and of prophenoloxidase (PPO) cascade, immune-responsive effectors, immune regulators, stress and oxidative stress transcripts (S1 Table). In addition, a manual annotation of a gene family involved in detoxification of xenobiotics–the cytochrome P450 (CYP)–was performed based on 143 CYPs of *Tr. castaneum* [38]. A total of 119 CYPs (76 full-length) and 49 fragments were annotated and four new sub-families discovered (CYP3160A, CYP3161A, CYP351E1 and CYP351F1) (S2 Table), further confirming the high quality of this transcriptomic database. In the absence of a genome of *T. molitor*, this transcriptome dataset allows identification of candidate genes from the present proteomic study, and is available to the scientific community for future proteomic or transcriptomic studies.

## Global proteome analysis reveals a restricted number of proteins transferred from mothers to the eggs

Analysis of the egg proteome by two-dimensional difference gel electrophoresis (2D-DiGE) allowed identifying 11 and 47 spots with a significant differential abundance of 1.5-fold between eggs from primed and control (PBS-injected) mothers in eggs sampled from ovaries or freshly laid (within 16 h after laying) 3 days post priming, respectively (Figs 1–2a and 2; S3 Table). While a special attention was paid during dissection and isolation of the eggs from ovaries to remove as much of mother's tissue as possible without impairing the eggs integrity, we cannot exclude that the eggs contained some mother tissue, notably nurse cells. The nurse cells are specialized cells known to be provisioning eggs with a wide variety of proteins, including immune effectors [39]. The higher number of proteins found differentially abundant in eggs collected from ovaries might at least in part be the consequence of nurse cells contamination of the sample. The fact that their differential abundance is significant and consistent over the six biological replicates however reflects that these differences are biologically relevant, at least at the level of the mother during egg development. Their absence in the eggs laid suggests that mothers control what is eventually transferred to them, which happens to be a restricted number of effectors from the 313 spots included in the proteome analysis (Fig 2, Table 2, S3 Table).

The Gram-positive bacterium *Bacillus thuringiensis* (*Bt*) induced the differential abundance of more proteins than the Gram-negative bacterium *Serratia entomophila* (*Se*) in eggs sampled in mother's ovaries (38 *vs* 22) (Fig 2, Table 2, S3 Table), which would be consistent with previous work showing that insects exhibit a higher induced persistent antibacterial response when primed with Gram-positive bacteria than with Gram-negative bacteria [19]. Sixteen were common between the two bacteria and they always exhibited the same pattern of over- or under-abundance (purple color in Fig 2, Table 2 and S3 Table), suggesting that most of the protein differences arise from a general defense mechanism against bacteria. Interestingly, this is no longer true for eggs laid, where most differences are found between the two bacterial priming (5, 3 and 4 spots for *Bt vs Se*, *Bt vs* Control and *Se vs* Control, respectively) while the same spots are not significantly differentially abundant when comparing either bacterial priming to the control (Table 2, S3 Table).

Proteins in all but one spots were successfully identified with a mean of 19 peptides covering 34% of target sequences, thanks to the high quality of the transcriptome assembly (S3 Table). Among the differentially abundant proteins, some are involved in response to stress (*e. g.*, Heat shock proteins HSP60, HSP70 and HSP75, desiccation stress protein), with notably the catalase which is known to protect cells from oxidative damage. Immune response to infection can generate an oxidative stress, which is handled by enzymes such as catalases that are generally over-expressed in response to infection [40,41]. Here, Catalase was consistently less abundant in eggs from mothers primed with *S. entomophila* than control (Table 2, S3 Table),

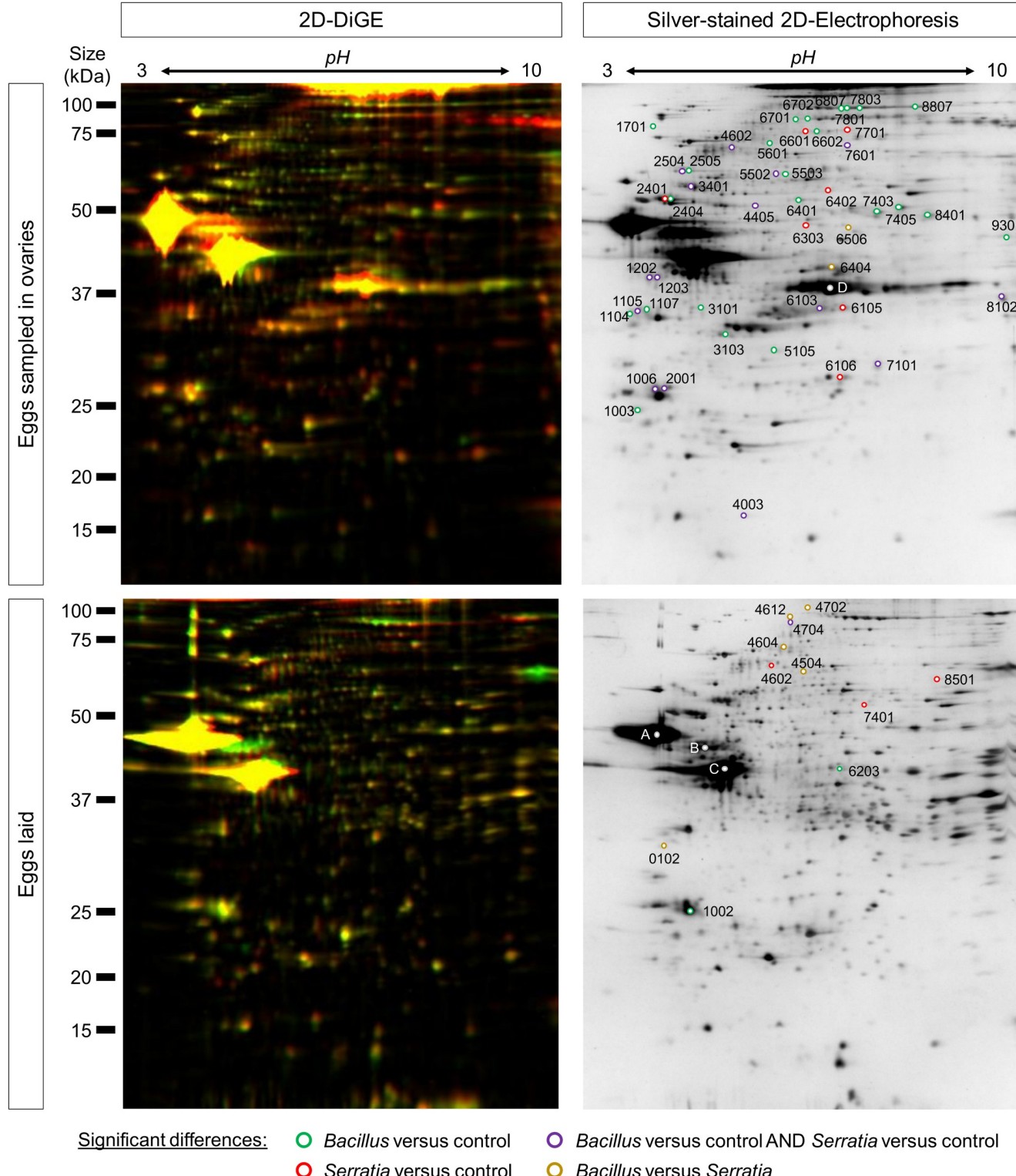

**Fig 2. 2D-gel electrophoresis highlights spots differentially abundant between priming conditions.** Left panels: two-dimensional difference gel electrophoresis (2D-DiGE) with the three fluorescent channels corresponding to the three CyDyes (Cy5, Cy3 and Cy2) merged. Right panels: 2D-SDS-PAGE stained by mass spectrometry-compatible silver staining protocol are shown for eggs sampled in ovaries (top gels) or sampled within 16 h post-laying (bottom gels). Eggs were collected 3 days after female priming. Spots significantly (p < 0.05, Mann-Whitney test) 1.5 differentially abundant in eggs from *B. thuringiensis* compared to control, *S. entomophila* compared to control, both *B. thuringiensis* and *S. entomophila* compared to control and *B.*

*thuringiensis* compared to *S. entomophila* are represented by green, red, yellow and purple circles, respectively. The same color code is used in Table 2 presenting a summary of proteins identified. A full list of the proteins identified and associated statistics is available in S3 Table. Six biological replicates per condition, each containing 70–74 eggs, for a total of 433 eggs from 117 different females were included in the 2D-DiGE analysis.

both in ovaries and eggs laid, which appears counter-intuitive. This is in line with the global lower abundance of most stress enzymes described above in eggs from primed females than in control ones (Table 2, S3 Table), potentially indicating a trade-off with important immune-related transferred proteins. Some canonical immune proteins were also identified, such as an Annexin, a protein involved in apoptosis and in many anti-inflammatory and infection responses and that is found differentially abundant only between *Bt* and *Se* conditions, and Prophenoloxidase, whose activation cascade is central in the arthropod innate immune response to bacterial infection through melanization [42]. Intriguingly, this enzyme was consistently less abundant in bacteria-primed females' eggs both in ovaries and laid (Table 2, S3 Table), suggesting that the protein itself might not provide the observed egg antibacterial protection.

Among the most abundant proteins in eggs from bacteria-primed females compared to control, some non-canonical immune proteins could be identified. A Transferrin was found more abundant in two spots (#6701 and #6702) in eggs from *Bt*-primed mothers (Fig 2, Table 2, S3 Table). Transferrins have been involved in immune response in vertebrates and invertebrates, notably thanks to their capacity to bind iron, thus creating an environment low in free iron that impedes bacterial survival in a process called iron withholding [43,44]. Two spots (#1006 and #2001) identified as Perilipin were also more abundant in eggs from both *Bt*- and *Se*-primed mothers than in control ones (Fig 2, Table 2, S3 Table). Perilipins are major constituents of lipid droplets that are dynamic organelles with a large range of important roles in cells, including lipid metabolism regulation, cell signaling, membrane trafficking and inflammation. They have notably been shown to participate in fundamental mechanisms of host-pathogens interactions, including cell signaling and immunity [45,46]. Transferrin and Perilipin are therefore good candidates for further investigation of their role in egg protection.

Many of the spots found differentially abundant between conditions were identified as Vitellogenin. It is the major nutrient source provided by the mothers into the eggs for the embryo development and is therefore expected to be highly abundant in eggs. However, Vitellogenin is more than a simple nutritive protein and has many additional roles including protection against oxidative stress [47,48] and antibacterial activities [49,50]. The latter is ensured by Vitellogenin fragments, which correspond to the spots observed on the gel that are at a

**Table 2. 2D-DiGE identifies differentially abundant proteins between the three conditions tested.**

| | Eggs laid | Eggs in ovaries |
|---|---|---|
| Bacteria > Control | *26S protease regulatory subunit*; <u>Gephyrin</u> | **Vitellogenin; Perilipin; Thioredoxin; Tubulin; Cofilin; Fascin-like; Elongation factor; HSP60; ruvB-like helicase;** *HSP75; Transferrin; Enolase; S-formylglutathione hydrolase; Dihydrolipoyl dehydrogenase; Imaginal disc growth factor* |
| Control > Bacteria | **Nuclear pore complex protein Nup93;** *Desiccation stress protein;* <u>Catalase</u>; Prophenoloxidase | **Vitellogenin; Prophenoloxidase; Aldose reductase;** *Annexin B9; Farnesoic acid 0-methyl transferase; Serine protease inhibitor;* <u>HSP70</u>; <u>Catalase</u> |
| *Bt > Se* | Annexin B9; Fascin-like; Prolyl endopeptidase | Vitellogenin; Enolase |
| *Se > Bt* | Vitellogenin; Gephyrin | Unidentified protein |

Proteins significantly (p < 0.05, Mann-Whitney test) 1.5 differentially abundant in eggs from *B. thuringiensis* compared to control, *S. entomophila* compared to control, both *B. thuringiensis* and *S. entomophila* compared to control and *B. thuringiensis* compared to *S. entomophila* are italicized, underlined, in normal case and in bold, respectively.

lower size than expected for the full protein (Fig 2, S3 Table). Vitellogenin has also been suspected to translocate bacterial proteins from mother's gut to the eggs to prime the embryo immune system as part of TGIP phenomenon [21,25,29]. In our proteome analysis, it appears difficult to conclude in the involvement of Vitellogenin for two reasons. First, the eighteen Vitellogenin spots exhibited various patterns of up- and down-regulation and were mostly observed in eggs from ovaries (S3 Table). Moreover, these spots represent a small fraction of the total Vitellogenin present in the eggs. Indeed, we aimed at identifying the nature of the most abundant spots seen in the egg proteome profile. These are the three spots present in both eggs laid and ovaries (named A, B and C), and one additional (named D) specific to eggs from ovaries (Fig 2, S3 Table). These four spots were all reliably identified as Vitellogenin (S3 Table). Unfortunately, due to their very high abundance, they could not be included in the analysis because their fluorescence intensity was saturated. Therefore, there might be differences in Vitellogenin abundance of even higher order between conditions that we could not observe due to limitations of the method used. In regard of the literature and based on these observations, Vitellogenin remains a good candidate for further investigation.

## Key antimicrobial peptides (AMPs) are specifically stored in eggs upon mother bacterial priming

Several AMPs have been involved in both within- and trans-generational immune protection [6,51,52]. The lower size range limitation of 2D gels excludes possible AMPs detection by such an approach. We therefore characterized the peptide profiles of eggs freshly laid (within 16 h after laying) 3 days post-priming by bacteria-primed and control females by a top-down proteomics approach, representing a total of 226 eggs from 54 different females. Using the above-described annotated *T. molitor* transcriptomic database, a database of *T. molitor* protein sequences was established and used for matching the MS/MS spectra observed by the top-down proteomics analysis of eggs extracts, resulting in more than twelve thousands identifications across the five samples (S4 Table).

Furthermore, a short list of fifteen known or candidate AMPs was extracted from this database (Table 3, S5 Table). *T. molitor* contains four AMPs named Tenecins that have been isolated and characterized for their immune function [30]. Tenecins 1, 2 and 4 are inducible antibacterial Defensin, Coleoptericin, and Attacin, respectively [33,53–56]. A previous study using Acid Urea-Polyacrylamide Gel Electrophoresis (AU-PAGE) and Tricine-SDS-PAGE analyses identified Tenecin-1 in eggs from *T. molitor* mothers primed with the Gram-negative bacteria *S. entomophila* and *Escherichia coli* and the Gram-positive bacteria *B. thuringiensis* and *Arthrobacter globiformis* but not in control ones [16]. Here, we confirm these results by identifying Tenecin-1 in eggs from *Bt* and *Se*-primed mothers but not from control ones (Table 3), supporting the validity of our approach.

Tenecins 2 and 4 were not detected in eggs from primed mothers in the above-cited previous study [16] but our sensitive top-down proteomics approach allowed detecting them (Table 3, S5 Table). Together with other Coleoptericins (c102099_g2_i1:1:417:1) and Attacins (c94080_g1_i1:1:529:2), Tenecins 2 (c101868_g1_i1:29:412:2 and c101868_g1_i2:29:412:2) and 4 (c104011_g1_i1:1:523:2 and c104011_g1_i2:1:523:2) were found in both types of samples albeit often with much higher identification scores in primed samples (Table 3, S5 Table). In the case of these Attacin proteins and Coleoptericin-A (c102099_g2_i1:1:417:1), the high difference in scores between samples could be associated with a difference in abundance. However, the low score differences observed for Tenecin 2 is rather indicative of similar levels between sample types. Altogether, this advocates that antibacterial protection of eggs not only relies on Tenecin-1 but might also depend on the presence of a wider range of AMPs, as it was

**Table 3. Top-down nano-LC-MS/MS identifies key candidate AMPs differing between the three conditions tested in freshly laid eggs (sampled within 16 h after laying) 3-day post-priming.**

| Transcript name | Tenecin name | AMP category | Sequest identification scores | | |
| --- | --- | --- | --- | --- | --- |
| | | | Control[1] | *B. thuringiensis*[1] | *S. entomophila*[1] |
| CL4266Contig1:2797:3051:1 | Tenecin 1 | Defensin | N/A; 18.00 | **68.89; 60.35** | **22.76** |
| c102099_g2_i1:1:417:1 | | Coleoptericin-A | N/A; 4.68 | **45.16; 92.81** | **32.74** |
| c101868_g1_i1:29:412:2 | Tenecin 2 (partial) | Coleoptericin-B | 12.81; 3.42 | **120.25; 136.34** | **38.06** |
| c101868_g1_i2:29:412:2 | Tenecin 2 (partial) | Coleoptericin-B | **61.82; 14.37** | **136.28; 137.15** | **41.01** |
| c94473_g1_i1:1:327:1 | Tenecin 3 | Thaumatin | **153.14; 718.92** | **99.13; 1482.00** | **35.08** |
| c104011_g1_i1:1:523:2 | Tenecin 4 (partial) | Attacin | N/A; 8.30 | **141.72; 202.48** | 9.57 |
| c104011_g1_i2:1:523:2 | Tenecin 4 (partial) | Attacin | N/A; 8.30 | **178.83; 240.75** | **45.68** |
| c94080_g1_i1:1:529:2 | | Attacin-2 | **24.71; 15.39** | **290.52; 268.59** | **130.86** |
| c94342_g1_i1:1:742:2 | | Thaumatin | 13.19; 18.67 | 4.44; 3.72 | N/A |
| c97779_g1_i1:1:778:2 | | Thaumatin | 1.91; 2.66 | 2.63; N/A | N/A |
| c98154_g1_i1:94:762:1 | | Thaumatin | N/A; N/A | 4.33; 1.45 | 3.8 |
| c93017_g1_i1:221:619:2 | | Thaumatin | N/A; N/A | N/A; N/A | N/A |
| c82414_g1_i1:1:233:3 | | Thaumatin-1 | N/A; N/A | N/A; N/A | N/A |
| c101919_g1_i1:1:491:3 | | Attacin-C | N/A; N/A | N/A; N/A | N/A |
| c101352_g1_i1 743:835:2 | | Cecropin | N/A; N/A | N/A; N/A | N/A |

1 High scores indicate both a reliable identification and an abundance of matching peptides while low scores rather suggest a weak identification of the protein and/or a low abundance in the sample. For readability reasons, only scores higher than 20 are in bold but all scores (low and high) are considered and discussed in the text. N/A: not found.

previously suggested [32,57]. The role in TGIP of all these AMPs should therefore be further validated.

In contrast with Tenecins 1, 2 and 4, Tenecin-3 is an antifungal peptide belonging to the thaumatin family that is constitutively expressed [58,59]. Our results established that Tenecin-3 is present in both replicates from control and eggs from *Bt*- and *Se*-primed mothers (Table 3). Tenecin-3 is the only thaumatin found in the *T. molitor* transcriptome that is detected in the eggs peptidome while the others were either not found or found with very low identification scores (Table 3, S5 Table). Tenecin-3 may have a housekeeping role in eggs to protect them from general fungal infections, irrespective from a bacterial exposure experienced by its mother.

## Interference RNA suggests that AMPs are actively transferred by mothers to protect eggs

To further investigate the direct transmission of antimicrobial immune effectors into eggs from primed mothers, we conducted a RNA interference approach aimed at comparing the antimicrobial protection of eggs from primed females (3–4 days old eggs laid 8-day post-priming) that were injected twice (3-day before priming and 1-day post-priming) with either i) PBS used as control of injection procedure, ii) dsRNA from GFP as a control for the effect of non-relevant dsRNA, or iii) a cocktail of dsRNA from various AMPs (Tenecin 1 (Defensin), Tenecin 2 (Coleoptericin B), Coleoptericin A, Tenecin 4 (Attacin) and Attacin 2). In this experiment, eggs did not show significant differences in antimicrobial activity, when measured as the size of the zone of inhibition (Fig 3A), but showed a significant difference when the proportion of eggs exhibiting an antimicrobial activity was measured. This proportion of protected eggs significantly decreased from 36.5% in eggs recovered from PBS-injected primed

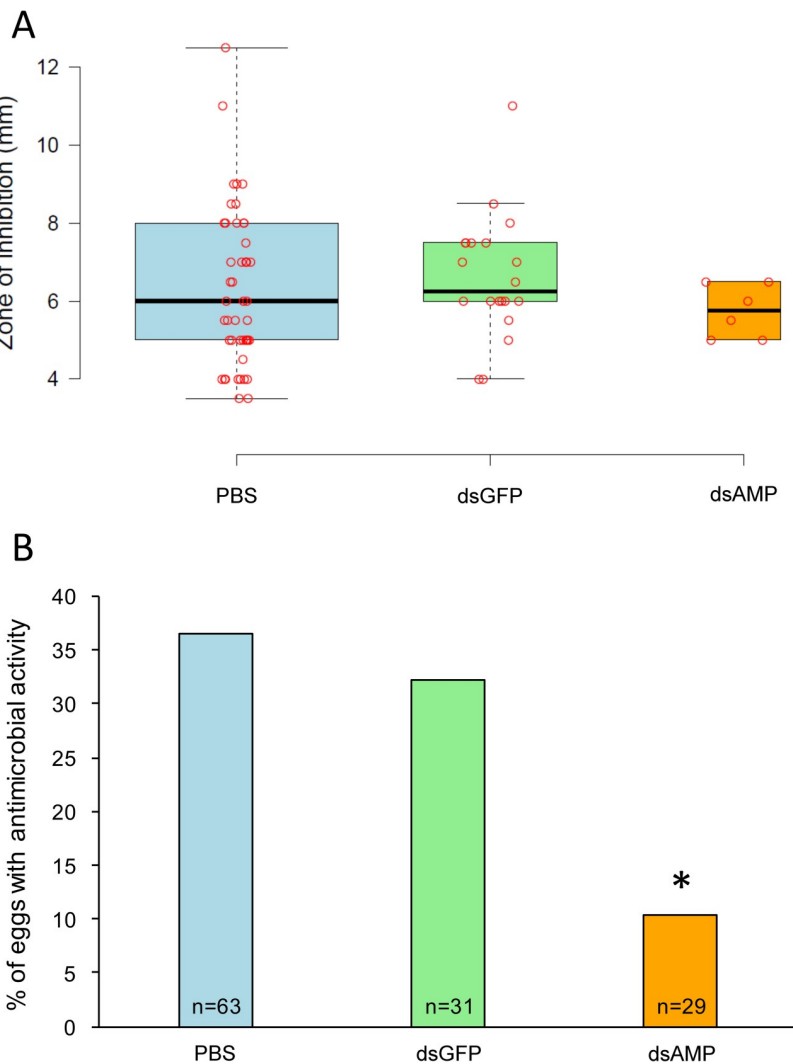

**Fig 3. AMPs condition the number of eggs protected but not the level of protection of protected eggs.**
Antimicrobial activity was measured in 3 days old eggs originated from females that were primed and injected with
either PBS (PBS), dsRNA from non-relevant GFP (dsGFP) or dsRNA targeting the five candidate AMPs (dsAMP).
Eggs were laid 5 days after priming. A. The zone of inhibition around eggs disposed on *A. globiformis*-covered agar
plates was measured. Eggs that did not exhibit antimicrobial activity (null zone of inhibition) were not considered. B.
The percentage of eggs exhibiting antibacterial activity is represented for the same three samples. A star '*' above the
bar indicates a significant difference between the dsAMP condition and the PBS and dsGFP samples ($p < 0.05$; Fisher's
exact test).

females to 10.3% in eggs originated from dsAMP-injected primed females (Fisher exact test,
$p = 0.012$) (Fig 3B). The latter matches the basal level of protection (~7–8%) of 3-day old eggs
laid by PBS-injected unprimed mothers previously reported in [19]. In the same time, the
injection of dsGFP had no effect on the proportion of eggs protected (Fisher exact test,
$p = 0.819$), highlighting that the observed effect is not procedural but biologically relevant
(Fig 3B).

Important individual variations were observed during all RT-qPCR performed to monitor
AMP candidate gene expression in RNAi-treated mothers (S1 Fig). Considerable variations in
response to dsRNA has previously been reported in insects, including coleopteran, and has
been linked to multiple contributing factors including dsRNA instability/degradation, delivery

method, dsRNA uptake efficiency, etc. [60,61]. In addition to a variation in RNAi efficiency (*i. e.* variation in silencing effect), it is possible that dsRNA (which mimics a viral infection) induces a variable non-specific immune response involving an increase in AMP expression. This would be consistent with the fact that we observed substantial variations in AMP expression in both females exposed to AMP ds-RNA and exposed to non-relevant GFP dsRNA. Despite this variability in AMP expression upon dsRNA treatment, our results show that eggs from primed GFP-dsRNA injected mothers are as likely to be protected as eggs from primed PBS-injected mothers, while eggs from primed AMP-dsRNA injected mothers are significantly less likely to display antibacterial activity. Therefore, even in the worst-case scenario where silencing of AMPs is limited (*i.e.*, differentially affecting the target AMPs and/or only partially reducing the AMPs expression), we would only underestimate the effect of AMPs on the TGIP phenotype observed, even if we cannot discriminate the effect of each specific AMP independently. Although we could not confirm AMP invalidation by RT-qPCR in RNAi-treated mothers and we did not investigate the antimicrobial activity in mothers' hemolymph, these results still provide support for an active transfer of AMPs as effectors by mothers into the eggs, enabling an efficient protection of a fraction of mature eggs (3–4 days old) laid by mothers 8 days after their immune priming.

## Mothers do not transfer transcripts but they boost candidate gene expression in mature eggs

Eleven candidate genes, respectively encoding four proteins (Transferrin, Perilipin, Prophenoloxidase and Vitellogenin) and seven AMPs (Tenecins 1–4, Attacin-2, Coleoptericin-A and Cecropin) identified by 2D-DiGE and MS/MS approaches, were selected to monitor the kinetics of their expression in bacteria-primed females and in their eggs (Fig 1-4). The objective was to determine whether mothers could transfer mRNA encoding these candidate proteins directly into the eggs and if their expression in eggs could be stimulated and/or synchronized with their expression in mothers.

One day after priming, only few significant although little differences in gene expression could be observed between females primed with bacteria and the PBS-injected control females (Fig 4A). However, bacterial exposure induced a much stronger increase in expression at day 5 post-priming of most AMPs (*attacin-2*, *coleoptericin-A* and *tenecins 1*, *2* and *4*), which were 3 to 20-fold significantly more expressed than in PBS-injected females. The maximum increase was observed for *attacin-2* (>20-fold) upon female priming with *B. thuringiensis* (Fig 4B). While over-expression of *tenecin-1* and *attacin-2* was maintained at day 12 post-priming–yet at a lower level than at day 5 –expression of other candidate genes was mostly back to ground level (Fig 4C). Only *cecropin*, *transferrin* and *vitellogenin* exhibited an increased expression upon priming with *S. entomophila*, while they were either not affected or slightly under-expressed at day 1 and 5. This delay of a few days between bacterial priming and expression in females of immune genes, and most notably AMPs, is congruent with other studies [33,56] and it corresponds to the increase in their antibacterial activity, which increases to reach its maximum after 2 days and slowly decreases to get back to the level of naïve females after 10 days [17].

Expression of candidate genes was measured at the same time-points considering that a previous work showed that the level of protection was low for eggs laid 1 day after female priming, then increased to its maximum between 3 and 8 days, to drop back to its ground level after 12 days [17]. Moreover, it was previously shown that egg antibacterial activity was strongly affected by its maturity, prompting us to test three egg ages, viz. 1, 3 and 7 days after having been laid. Interestingly, although the expression of the housekeeping genes could be detected

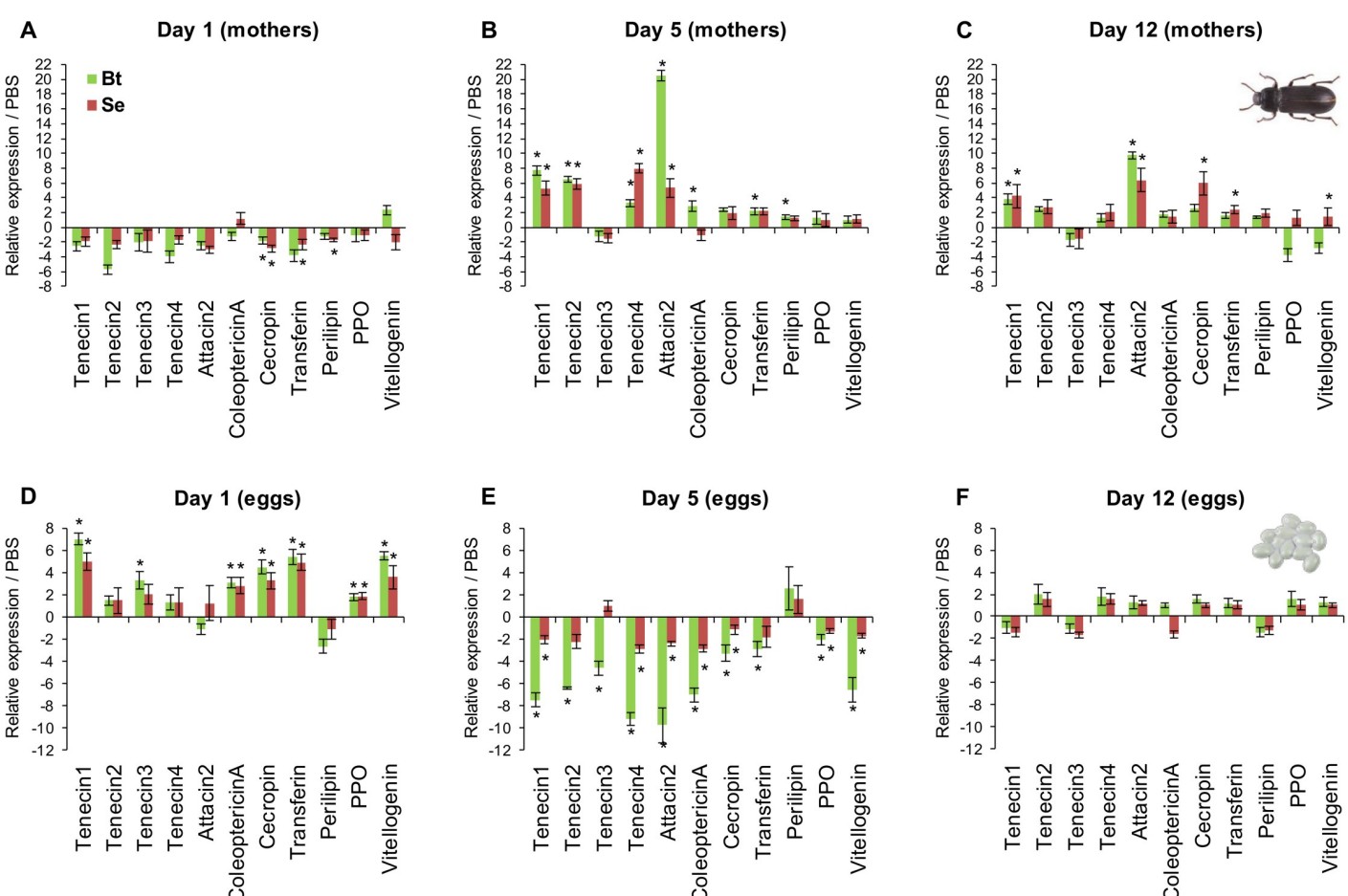

**Fig 4. Mothers and eggs exhibit desynchronized candidate gene expression patterns.** Expression of the 11 candidate immune genes was measured in adult females (panels A to C) and in 7 days-old eggs (panels D to F) from mothers 1, 5 and 12 days post-priming (panels A and D, B and E, C and F, respectively). Values related to priming with *B. thuringiensis* and *S. entomophila* are represented by green and red bars, respectively. Data are represented as mean expression in primed condition relative to PBS-injected control. A '*' above the bar indicates a significant over- or under-expression in primed condition compared to control (Mann-Whitney test, $p<0.05$).

in all replicates from all priming conditions (1, 5 and 12 days post-priming) and all egg ages (1, 3 and 7 days old), the level of expression of immune genes was below the detection threshold for ~90% and ~70% of them for 1 day-old and 3 days-old eggs, respectively. Considering that we observed a high fluctuation between replicates for the same genes (*i.e.*, just above the detection threshold for some and undetectable for others), no reliable quantitative data could be obtained for these two egg ages. These results highlight two phenomena. First, females did not transfer mRNA of these eleven candidate genes at a level sufficiently high to be reproductively detected by RT-qPCR, suggesting that this does not partake to egg effector production and immune protection. Second, it shows that the transcription machinery of young eggs is able to express housekeeping genes that are mandatory for the proper embryo development but that it is too immature to express other genes, especially immune-related ones. In insect eggs, the serosa is an extraembryonic membrane enfolding the embryo that ensures its protection against desiccation and infection, notably by expressing the genes encoding important immune effectors [62–64]. Considering that the serosa takes approximately 3 days to be established and functional in *T. molitor* [32,65], this might explain why a limited and low candidate gene expression was found for eggs younger than 3 days old in our RT-qPCR analysis.

Seven days-old eggs laid by females 1 day after their bacterial priming exhibited a significant over-expression (2- to 7-fold) of most of their immune genes, *i.e. tenecin 1* and *3, coleoptericin-A, cecropin, transferrin, prophenoloxidase* and *vitellogenin* (Fig 4D). At 5-day post-priming, the expression of all genes except *perilipin* was significantly down-regulated from 2 to 10-fold in eggs from primed females compared to eggs from PBS-injected female (Fig 4E). At 12 days post-priming, the expression of all candidate immune genes was back to ground level (Fig 4F). Expression pattern of candidate genes in eggs is therefore a transient phenomenon, with a global increase, then decrease to get back to normal level along the post-priming days while females at the same days post-priming exhibit different pattern, showing its highest levels of expression at day 5 (Fig 4).

## Antibacterial activity of eggs supports that their protection relies on maternal transfer of effectors rather than transcripts

Eggs laid by naive (negative control) females and by females injected with PBS (control injection), *B. thuringiensis* or *S. entomophila* were collected 1 day and 5 days after priming and left for maturation for 7 days before testing their antibacterial activity. The proportion of naive and PBS-injected females laying eggs with antibacterial activity was similar, whether eggs were sampled at 1 or 5 days post-priming (logistic regression: maternal priming $\chi^2 = 2.09$ df = 1 $p = 0.149$, day post-priming $\chi^2 = 0$ df = 1 $p = 0.999$, Fig 5). One day after priming, the proportion of females laying eggs with antibacterial activity was low and similar among the four different immune treatments (logistic regression: $\chi^2 = 0.93$; df = 3; $p = 0.818$). It is five days after priming that the proportion of females laying eggs with antibacterial activity show significant variation among immune treatments (logistic regression: $\chi^2 = 16.75$, df = 3, $p = 0.001$). In more details, while *B. thuringiensis* and *S. entomophila* each induced a significant difference in the proportion of females protecting eggs (*Bt vs* naive: $\chi^2 = 7.47$, df = 1, $p = 0.006$; *Se vs* naïve: $\chi^2 = 11.15$, df = 1, $p = 0.001$), the proportion of females protecting eggs between the two bacterial treatments did not significantly differ (*Bt vs Se*: $\chi^2 = 1.50$, df = 1, $p = 0.220$). Taken together with the RT-qPCR results, these antibacterial activity data show that the over-expression of candidate genes in mature eggs (7 days-old) from 1 day-primed females does not induce an increased protection, which is consistent with previous results on *T. molitor* showing that LPS-primed females start protecting eggs from day 3 to day 8 post-priming [17]. This over-expression therefore does not seem to lead to a sufficiently high amount of effectors to permit a significantly higher protection of the eggs. However, the egg protection observed herein matches with the high expression level of candidate genes in 5 days-primed females, suggesting that females might directly transfer effectors to their eggs, protecting them as early as 1 day after they are laid until 7 days post-laying [19].

## Disentangling the different scenarios of TGIP in *T. molitor*

Different scenarios were recently proposed as a framework to identify the different mechanisms that could be at play to support TGIP in invertebrates [6]. Here we present the first empirical study aiming at unraveling whether mothers transfer a "signal" (scenario 1), mRNA (scenario 2) and/or protein/peptide effectors (scenario 3) to protect their offspring at the egg stage, taking the mealworm beetle *T. molitor* as the model organism.

Our global proteomic approach allowed investigating if the transfer of effectors (scenario 3) occurred by studying eggs freshly laid or collected directly from ovaries. As discussed previously, we revealed that mothers were able to transfer different proteins and peptides, including notably many different AMPs that were responsible for the increased egg immune protection observed in mature eggs. In addition, we managed to determine whether the transfer of

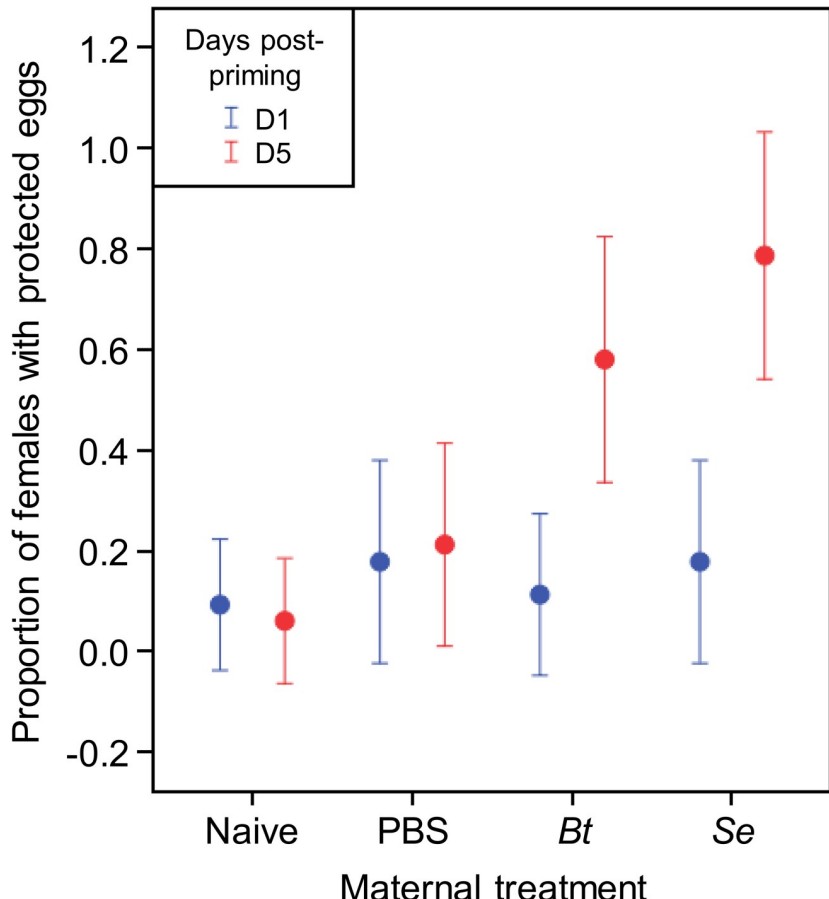

**Fig 5. Mothers protect their eggs at 5 days–but not 1 day–post bacterial priming.** Proportion of females with protected eggs in function of the maternal treatment. Data on 7-day old eggs laid by females after 1 day and 5 days post-priming are represented in blue and red, respectively. Eggs (3 per female) collected at day 1 were from 22 naïve females, 17 PBS, 18 *Bt* and 17 *Se*. Eggs collected at day 5 were from 17 naïve females, 19 PBS, 19 *Bt* and 14 *Se*. A total of 321 eggs were included in the analysis. Error = confidence interval 95% (logistic regression).

effectors was an active phenomenon (*i.e.*, mothers controlling the quantity of effectors transmitted and the number of eggs concerned) or exclusively passive (*i.e.*, effectors circulating in mother's hemolymph ending up in eggs by diffusion phenomenon). If mothers passively transferred effectors to the eggs, one would have expected that all eggs would be protected at the same level, established by a fine equilibrium between the concentration of effector in the circulating system of the mother and in the eggs. Upon inhibition of the expression of candidate AMPs in the mothers by treatment with dsRNAs, one would therefore have expected a drastic drop in the overall protection of the eggs but all eggs should have exhibited a similar low protection. In contrast, we observed that the number of eggs protected significantly decreased but the level of protection of the protected eggs remained as high as eggs from primed mothers untreated with AMP-specific dsRNA. One should be careful that the importance of AMP involvement in TGIP could even be stronger than observed in the present study for *T. molitor*, as we might have observed an attenuated phenotype given our inability to validate the knockdown of the AMPs in mothers. This nevertheless supports that mothers actively protect eggs in a binary way: either providing them with a sufficient amount of effectors to efficiently protect them against early bacterial aggressors, or not providing any protection, leading to no-to-low

antibacterial activity. Such active transfer strategy is evolutionarily sound, giving a higher chance of survival to a subgroup of eggs rather than risking the loss of the entire offspring population in case of massive infection if the transfer was exclusively passive. Particularly, since TGIP of the eggs was found to be a costly process for *T. molitor* females [17,31], females may take advantage of adjusting the proportion of eggs they may protect according to their individual perception of the risk of dying from an infection and the expected parasitic conditions for the offspring. That a part of the transfer is passive cannot however be deciphered based on the present data and will require further specific investigation.

We demonstrated that the level of immune gene transcripts in young eggs (1 and 3 days old) was too low to be detected by RT-qPCR approach. Considering that the RT-qPCR is a highly sensitive method generally able to detect as low as ~10–50 transcript copies per µg of RNA [66,67], this could suggest that the low expression level in young eggs might not be sufficient to significantly affect the amount of effectors. Therefore, if mothers indeed transferred mRNA into eggs, its effect might be marginal. However, eggs increased the expression of immune genes after only 1 day post female priming while mothers only reacted 5 days after. This suggests that eggs are able to mount a transcriptional response to maternal infection only when they are mature (*i.e.*, when their serosa is fully formed as discussed above) and when they are laid promptly after mother's priming. This would support the transfer of a 'signal' that would boost the expression of immune genes in the eggs while mothers are not expressing them yet. The question of the nature of such signal has been investigated during the last few years. Some authors argued that females of *T. castaneum*, *Apis mellifera*, *Galleria mellonella* and *Manduca sexta* could transfer bacterial peptides from their gut into the eggs, providing the embryo with antigens from pathogens to which they could be exposed [21,25,29,68]. However, it does not seem to be a universal process as it was not evidenced in *M. sexta* females exposed to the Gram-negative bacterium *Serratia marescens* [11]. Some even provided evidence that these bacterial peptides are translocated by vitellogenin during its storage into *A. mellifera* eggs [25]. While we found differential gene expression and protein abundance of vitellogenin in our analyses, our attempts to detect bacterial peptides transfer from *T. molitor* mother's gut to the eggs, based on the previously published protocol on *T. castaneum* from Knorr et al. [21], were inconclusive. Additionally, the proteomic and peptidomic analyses done during this study did not isolate any bacterial protein fragment. Altogether, while these data do not firmly exclude that such phenomenon is involved, they definitely call for future studies to optimize the protocols to monitor such phenomenon and confirm if and when it might occur (*i.e.*, is it pathogen and/or insect specific? What is the timing between priming and monitoring at which it occurs?). Even if such a 'signal' transfer does exist, the increase in immune gene expression in mature eggs 1 day post maternal priming might be transient and/or too low to produce a sufficiently high quantity of effectors to efficiently protect the eggs, considering that the proportion of 7-day old eggs from 1-day primed females exhibiting an antibacterial activity did not increase. When eggs were laid later after mother's immune priming, the transfer of effectors might have been sufficient to sustain the antibacterial protection of eggs and even downregulate the corresponding genes.

Another possibility is that mothers transfer their immunological experience by epigenetic modifications (scenario 4 in [6]), as it has been recently proposed and demonstrated in *M. sexta* [68]. This could be an additional explanation for the increased gene expression of mature eggs laid by females 1 day after bacterial priming. However, the present study did not intend to explore such scenario, which requires a comprehensive dedicated approach to provide reliable and conclusive data, notably to characterize gene methylation and histone acetylation throughout the development of the insect and during successive generations. Such studies remain rare but could in the future complement our present work aiming at disentangling the different

TGIP scenarios in order to determine if epigenetic is supporting, at least in part, TGIP in *T. molitor*.

## Material & methods

### Insect cultures, bacterial growth and procedure of females infection

All insects used in this study were from stock cultures maintained in standard laboratory conditions (24 ± 2˚C, 70% RH in permanent darkness). They were provided *ad libitum* with bran flour, water and supplemented once a week by apple.

Females were primed with the Gram-positive *Bacillus thuringiensis* (*Bt*) and the Gram-negative *Serratia entomophila* (*Se*). These bacteria are known natural pathogens of *T. molitor* [30] and they induce contrasting immune priming responses within and between generations in this insect [15,16,19]. The bacteria were all obtained from the Pasteur Institute (*Bt*: CIP53.1; *Se*: CIP102919) and suspensions for immune priming were prepared as described in [15]. Briefly, the bacteria were grown overnight at 28˚C in liquid broth medium (10 g bacto-tryptone, 5 g yeast extract, 10 g NaCl in 1000 mL of distilled water, pH 7). They were then inactivated in 0.5% formaldehyde prepared in PBS for 30 min and rinsed three times in PBS. Inactivation was tested by plating a sample of the bacterial solution on sterile broth medium with 1% of bacterial agar and incubated at 28˚C for 24 hr. Aliquots were kept at −20˚C until use.

For all experiments, immune priming was performed on virgin females (10 ± 2 days post-emergence) by first chilling them on ice for 10 min for immobilization purpose and then by injecting a 5-μL suspension of inactivated bacteria (*Bt* or *Se*) or of buffer only (procedural control for effect of the injection). Injections were done through the pleural membrane between the second and third abdominal segments using sterile glass capillaries that had been pulled out to a fine point with an electrode puller (Narashige PC-10). For both injected bacteria, the concentration was adjusted to $10^8$ microorganisms per mL using a Neubauer improved cell counting chamber [16]. Immediately after their treatment, the females were paired with a virgin and immunologically naive male of the same age and allowed to produce eggs in a Petri dish supplied with wheat flour, apple and water in standard laboratory conditions (24˚C, 70% RH; dark).

### *De novo* assembly of *T. molitor* Illumina transcriptome (RNAseq)

In the absence of genome, a reference *T. molitor* transcriptomic database (Fig 1-1) was necessary to identify candidate proteins from the proteomic study. Because we aimed at generating a database enriched in transcripts involved in immune and stress responses, the sequencing was performed on pooled RNA from individuals of various developmental stages (third instar larvae, pupae, adults), sex (males and females) and physiological conditions. More precisely, groups of 10 individuals of third instar larvae, adult males or adult females, were either not treated or injected with *B. thuringiensis*, or with *S. entomophila* bacteria as described above, or injected with the drug phenobarbital (0.1%), known to induce a detoxification response in insects [69]. We also used RNA from 30 pupae (untreated because of their higher sensitivity to stress), resulting in a total of 150 individuals. RNA was extracted 48h after priming, using trizol method (TRIzol LS Reagent, Invitrogen) according to the manufacturer's instructions.

Sequencing was carried out on an Illumina HiSeq2000 Genome Analyser platform using paired-end (2x100bp) read technology with RNA fragmented to an average of 380 nucleotides. Sequencing of two technical replicates was performed by Eurofins-mwg-operon and resulted in a total of 70 and 51 million paired-end reads. Quality control measures, including the filtering of high-quality reads based on the quality score given in fastq files (FactQC, version0.10.1), removal of reads containing primer/adaptor sequences and trimming of read length, were carried

out using Trimmomatic (version 0.3.1 [70]). Reads with Phred-like score <20 and read length less than 40 nucleotides were removed. After this quality filtering, the two technical replicates resulted in a total of 58 and 44 million reads that were pooled to obtain a reference transcriptome. The *de novo* transcriptome assembly was carried out using Trinity (version 2014/09/07 [71]) with k-mers sized 25, T = 50 and Jaccard similarity coefficient (option from trinity to reduce chimeric transcripts). Our *de-novo* transcriptome contains 110,963 transcripts with an N50 (sequence length of the shortest contig at 50% of the total transcriptome length) of 1261 nucleotides and an ExN50 of 1135 nucleotides. We used Bowtie (version 0.12.9 [72]) to align reads in our transcriptome. Complete metrics can be found in Table 1. To reduce the numbers of transcripts we performed a super-assembly with TGICL (version 2.1 [73]) using default settings.

Translation of the transcriptome was performed using FrameDP (version 1.2.0 [74]) and Uniprot (uniprot.org, version of 29April2015) was used as database for the machine learning phase of detection of the best genetic code. The resulting predicted proteins (45,505) were compared using BLASTP (version 2.2.30+, e-value of $10^{-5}$) and *Tr. castaneum* proteome available at Uniprot (version 14April2015). Functional proteome annotation was predicted using InterProScan (version 5 [75]) on the GALAXY-BBRIC INRA platform (bbric.toulouse.inra.fr) using InterPro database (version 52). CEGMA analysis was used to validate the quality of the transcriptome-proteome [76]. Our reference transcriptome, its proteome and annotation were used as a resource for candidate genes and proteins in the following experiments. They are available for download on the IHPE laboratory website (http://ihpe.univ-perp.fr/acces-aux-donnees) and on the NCBI database under the BioProject ID PRJNA646689 with SRA numbers SRR12235350 and SRR12235349.

## Global egg proteome and AMPs analysis

Proteomic and peptidomic analyses were performed on eggs laid or extracted directly from ovaries from primed and control females. Two complementary experiments, 2D-DiGE (Fig 1-2a) and mass spectrometry (Fig 1-2b), were conducted to characterize the proteins and peptides differentially abundant between eggs originating from primed females and those from control ones, respectively.

Considering that *T. molitor* females were reported to protect their eggs through TGIP from day 2 to day 8 post-maternal priming [17], only eggs produced after the third day following the maternal immune treatment were collected (Fig 6A). Eggs were either collected directly in the ovaries or freshly laid in the Petri dish. In the latter case, laid eggs have always been collected within 16 hours after laying to make sure that the embryo has not started its development. To make this possible, couples were transferred into a new Petri dish supplied with fresh flour, apple and water three days after the maternal treatment. Eggs produced into this new Petri dish were collected 16 hours after couple was transferred (Fig 6A). At the moment of egg collection, females were chilled on ice for 10 min and then dissected in ice cold PBS to remove the ovaries. Isolated eggs from ovaries were then rinsed in clean cold PBS to removed small sticky ovarian tissue and then gently dried for few seconds on sterile filter paper tissue before collecting and immediately frozen in liquid nitrogen and stored at -80˚C until use. Females used to collect laid eggs were not dissected. Their eggs were sieved and treated as above.

A two-dimensional difference gel electrophoresis (2D-DiGE) approach was used to qualitatively and quantitatively determine the differential abundance of proteins (of size ranging from ~10 to 150 kDa) between conditions (eggs from primed females *vs* eggs from naïve females, eggs primed with *Bt vs* eggs primed with *Se*). 2D-DiGE uses direct labeling of proteins with fluorescent dyes (CyDyes: Cy2, Cy3, and Cy5) prior to their isoelectrofocusing to solve the known quantitation and reproducibility problem of 2D-Electrophoresis.

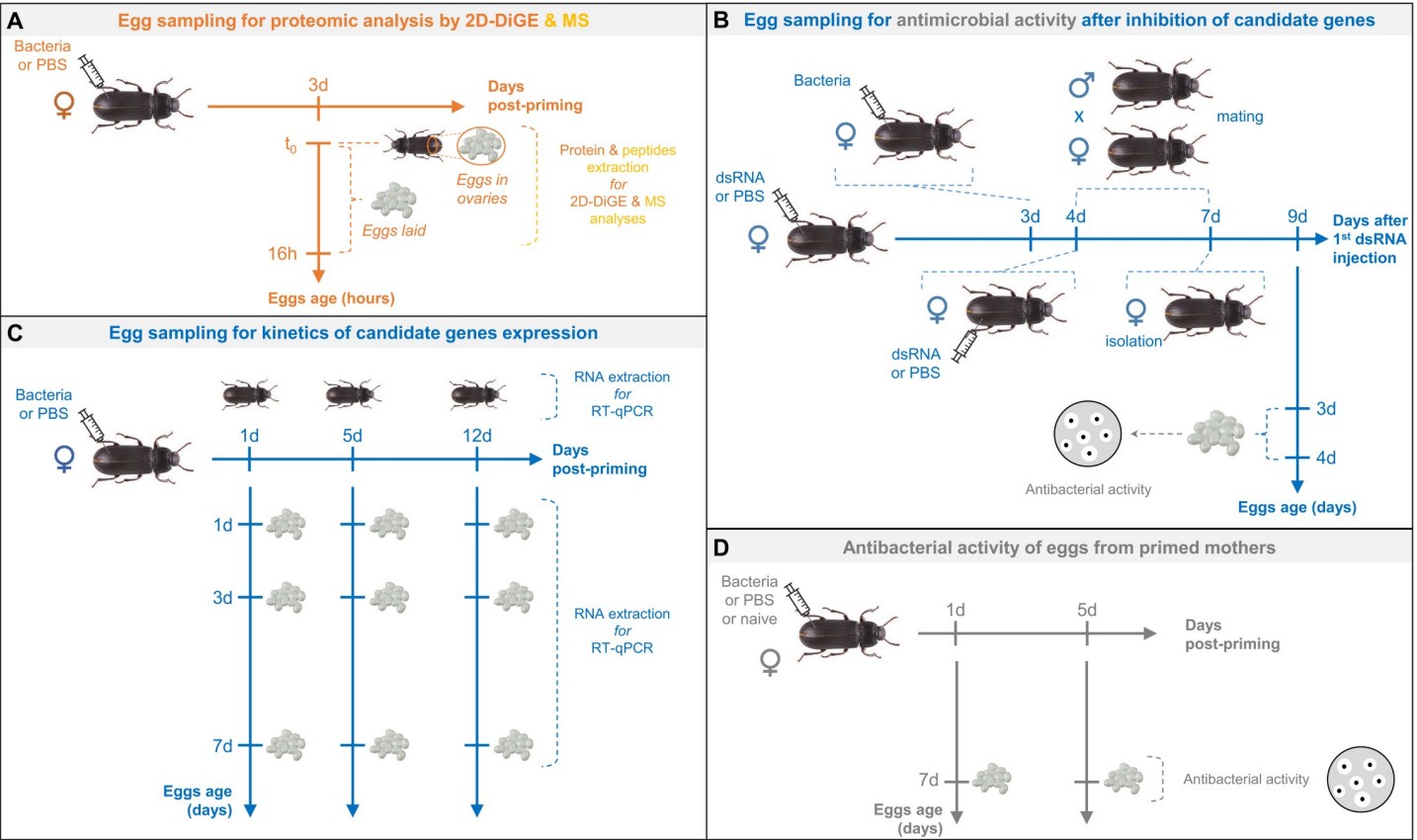

**Fig 6. Four complementary protocols for female adult priming and egg sampling were conducted to elucidate the different scenarios of TGIP in *T. molitor*.** Egg sampling strategies for proteomic approach (panel A), antimicrobial activity following RNAi experiment (panel B), kinetics of gene expression by RT-qPCR (panel C) and antibacterial activity following mothers priming (panel D) are indicating. A full description of the procedure for each experiment is described in details in the corresponding part of the Material and Methods section.

A total of 433 eggs were collected 3-day post-priming (70–74 per condition) from 117 different females (16–24 per condition) injected at 2 or 5–7 different dates for eggs in ovaries and freshly laid (sampled within 16 h post-laying), respectively. For each condition, eggs were pooled into 6 different biological replicates, each constituted of 11–17 eggs from 2–7 different females injected at the same date when possible. All information on female priming date, egg sampling and pooling are available in S6 Table. For each replicate, eggs were ground in UTTC buffer (urea, 7 M; thiourea, 2 M; Tris, 30 mM; CHAPS, 4%; pH 8.5) using a sterile pestle and incubated at RT for 2 hours. After centrifugation (5 min at 10,000 g), the supernatant was collected and the protein concentration was quantified using the 2D-Quant Kit (GE Healthcare) following manufacturer's instructions, before being stored at -80°C until use. Fifty micrograms of proteins were labeled with either Cy3 or Cy5 while 50 μg of an equimolar pool of proteins from all extracts were labeled with Cy2 as an internal standard. The CyDye minimal labeling of the purified proteins was performed following manufacturer's instructions (GE Healthcare). A dye swap was performed to ensure that the observed differences between the three different priming conditions for eggs laid and eggs in ovaries were not due to different labeling efficiencies of the dyes. The DiGE labeling setup is available in S6 Table. Labeled proteins were then mixed together and diluted to a final volume of 340 μL with rehydration buffer (urea, 7 M; thiourea, 2 M; CHAPS, 4%; DTT, 65 mM) containing 0.2% of Bio-Lyte 3/10 ampholyte (Bio-

Rad). Isoelectrofocusing was performed as previously described [77]. Briefly, sample was loaded on a 17 cm ReadyStrip IPG strip with a non-linear 3–10 pH gradient (Bio-Rad) for passive (5h) and active rehydration (14h at 50V). Focusing was performed using the following program: 50 V for 1 h, 250 V for 1 h, 8,000 V for 1 h, and a final step at 8,000 V for a total of 50,000 V.h with a slow ramping voltage (quadratically increasing voltage) at each step. Rehydration and focusing were both performed on a Protean IEF Cell system (Bio-Rad). After reduction with DTT and alkylation with iodoacetamide of the proteins, strips were loaded on top of a 12%/0.32% acrylamide/piperazine diacrylamide gel and run at 25 mA/gel for 30 min followed by 75 mA/gel for 8 h using a Protean II XL system (Bio-Rad). Protein standards (Unstained Precision Plus Protein Standards (Bio-Rad)) were loaded on Whatman papers disposed on the left part of gels. Gels were scanned using a ChemiDoc MP Imaging System (Bio-Rad) associated with Image Lab software version 4.0.1 (Bio-Rad) using the blue (530/28 filter), green (605/50 filter) and red (695/55 filter) epi-illumination parameters for scanning Cy2, Cy3 and Cy5-labeled proteins, respectively. The qualitative and quantitative comparative analysis of digitized proteome maps was conducted using the image analysis software PDQuest 7.4.0 (Bio-Rad). Only spots significantly ($p < 0.05$, Mann-Whitney test) 1.5-fold differentially abundant between conditions were considered. In order to identify the proteins in the different spots of interest, classical 2D-SDS-PAGE were conducted on each condition separately and gels were stained following a mass spectrometry-compatible silver staining procedure previously described in [77]. Spots were excised using a Onetouch Plus Spot Picker Disposable (Harvard Apparatus) equipped with specific 1.5-mm methanol-washed tips. For each spot, protein in gel plug was trypsin-digested and digested peptides were analyzed with a nano-LC1200 system coupled to a Q-TOF 6550 mass spectrometer equipped with a nanospray source and an HPLC-chip cube interface (Agilent Technologies) as previously described [77]. Protein identification was performed by extracting the peak lists and comparing with the *T. molitor* translated transcriptome database by using the PEAKS studio 7.5 proteomics workbench (Bioinformatics Solutions Inc., build 20150615). The searches were performed with the following specific parameters: enzyme specificity, trypsin; three missed cleavages permitted; fixed modification, carbamidomethylation (C); variable modifications, oxidation (M), pyro-glu from E and Q; monoisotopic; mass tolerance for precursor ions, 20 ppm; mass tolerance for fragment ions, 50 ppm; MS scan mode, quadrupole; and MS/MS scan mode, time of flight. Only significant hits with a false discovery rate (FDR $\leq$ 1%) for peptide and protein cutoff ($-logP \geq 13$ (corresponding to a p-value of 0.05) and number of unique peptides $\geq 3$) were considered for top hits. For ensuring a proper identification of the proteins, a BLAST search against NCBI nr database was performed and the conserved domains of the sequence were retrieved using the NCBI CD-search available at https://www.ncbi.nlm.nih.gov/Structure/cdd/wrpsb.cgi [78]. For each protein, pI and molecular mass were calculated with the ExPASy Compute pI/Mw tool (available at http://web.expasy.org/compute_pi).

A limit of the 2D-DiGE approach is its inability to reliably reveal small proteins (<10 kDa) and peptides, which include the AMPs whose role in TGIP has already been demonstrated [8,16,21,23,27,29,79,80]. We investigated their abundance in eggs freshly laid (sampled within 16 h post-laying) by mothers 3 days after *Bt* and *Se* priming compared to PBS-primed ones. Two replicates consisting of 51 and 31 eggs and 49 and 43 eggs were prepared for the control and *B. thuringiensis* conditions, respectively. Only 52 eggs were available from *S. entomophila*-primed mothers and they constituted the only replicate for this condition. Eggs originated from a total of 54 different females primed at 8 different dates (S6 Table). Due to the limited number of replicates, only qualitative differences were extrapolated from data to identify candidate proteins. We first incubated each egg pool in trifluoroacetic acid (TFA) 0.1% to perform an acidic extraction of peptides that were then purified using a Sep-Pak C$_{18}$ Plus Light

Cartridge (Waters) following manufacturer's instructions. Peptides were eluted in 5 mL acetonitrile 60% / TFA 0.1% and lyophilized overnight. Protein concentration was estimated by their absorbance at 205 nm using NanoDrop One Spectrophotometer (Thermo Fisher Scientific), indicating that we extracted approx. 60 μg of protein per egg sample. Peptides were then suspended in 50 μL of acetonitrile 2% / TFA 0.1% and their quality was first assessed by MALDI-TOF (Matrix Assisted Laser Desorption Ionization—Time of Flight) fingerprinting with the HCCA Biotyper matrix (Bruker Daltonic, Germany) mixed with three different dilutions for each sample. MALDI analysis was run on an Autoflex III Smartbeam (Bruker) controlled by the Bruker Compass 1.4 software. Once validated, samples were diluted in ammonium bicarbonate (ABC) 100 mM, reduced by adding DTT (11 mM final concentration, incubation 30 min at 56˚C), and alkylated with iodoacetamide (34 mM final, 1h at room temperature, in the dark). Samples were then acidified with diluted TFA, centrifuged 10 min 15,000 g at 4˚C, and the supernatant volume was reduced by speed-vacuum. The supernatants were then analyzed by top-down LC-MS/MS: one-tenth of each sample was injected on an Ultimate 3000 nano-HPLC system equipped with a 75-μm $C_{18}$ column and connected to a Q-Exactive Orbitrap high resolution mass spectrometer operated in data-dependent acquisition positive mode (all from Thermo Scientific, Germany). Components, solvents, and operating parameters for nano-LC-MS/MS analysis were as described by [81]. Spectra from the acquired MS/MS files were matched to the sequences of the database of *T. molitor* proteins we built for this study, using the software Proteome Discoverer (version 1.4) set with the following parameters: no enzyme and 12 / 144 amino acids as minimum / maximum peptide lengths, respectively, tolerance of 10 ppm / 0.02 Da for precursors and fragment ions, carbamidomethylation of cysteine set as a fixed modification, C-terminal protein amidation, and methionine and tryptophan oxidation set as variable modifications, analysis in high-confidence mode (false discovery rate 1%). The Sequest HT algorithm implemented in Proteome Discoverer software was used for the database search. A short list of fifteen known or candidate AMPs proteins was extracted from this database, by screening the annotations obtained with the transcriptomic database and by a literature search. High scores indicate both a reliable identification and an abundance of matching peptides, potentially reflecting a high abundance of the related protein. Low scores, usually below 20, rather suggest a weak identification of the protein and/or a low abundance in the sample. For readability reasons, only scores higher than 20 were highlighted in the Table 3 but all scores (low and high) are discussed. The mass spectrometry proteomics data have been deposited to the ProteomeXchange Consortium via the PRIDE partner repository with the dataset identifier PXD018772.

## Functional invalidation of candidate immune genes using RNA interference

This experiment aimed at testing the contribution of candidate AMP genes in egg protection through TGIP by knocking down their expression using RNAi technology (Fig 1-3). Despite extensive investigation efforts, important individual variations in AMPs expression levels prevented the observation of a firm effect of injection of a single AMP ds-RNA on the underexpression of AMPs in mothers. Efforts to reduce variability included using non-intrusive delivery method of dsRNA ingestion [60,61], testing the effect of a single or double exposure to dsRNA and testing several time points after exposure to dsRNA. None of these trials resulted in decreased variations in AMP expression levels in mothers and in a significant effect on eggs antimicrobial activity. Of note, it has been shown in *Drosophila* model that the knockdown of single TEP (Thioester-containing protein) genes with redundant functions was insufficient to significantly impact the insect phenotype and that a change in microbial infection

resistance was only observed upon the knock-down of all 4 TEP genes simultaneously [82]. Therefore, we decided to inject females with a cocktail of different candidate AMP genes to maximize the potential phenotype effect on the eggs. For this purpose, on day 1, 10-day post-emergence females were injected with 5 µL of either i) PBS as control for injection procedure, ii) GFP-dsRNA (0.6 µg/µL) as control for non-relevant dsRNA, or iii) a cocktail of dsRNA from *tenecins 1* and *4*, *coleoptericins* A and B, and *attacin-2* (0.12 µg/µL each, 0.6 µg/µL for total ds-RNA), after being chilled on ice for 10 min. Double-stranded RNA (dsRNA) were synthesized by *in vitro* transcription using the MEGAscript T7 kit (Ambion, UK) according to the manufacturer's recommendations. Regions of approximately 200 bp were synthesized from *tenecin-1* (GenBank accession number Q27023), *tenecin-4* (GenBank accession number AB669089), *coleoptericin-A* (GenBank accession number KF957599.1), *coleoptericin-B* (GenBank accession number KF957600.1), and *attacin-2* (GenBank accession number MF754108.1). Non-relevant dsRNA used as a negative control consisted in a 208 bp portion of the jellyfish green fluorescent protein (*gfp5*) RNA (GenBank Accession number L29345).

Females were individually placed in a 12-well plate, containing bran flour and a 10 µL drop of gelose for hydration (1% agar-agar, 10% sucrose) until day 3. At day 3, all females were immune primed by injection of a 5 µL suspension of inactivated *B. thuringiensis* ($10^8$ bacteria/mL) as described above (Fig 6B). On day 4, females were injected again with either i) PBS as control for injection procedure, ii) GFP-dsRNA (0.6 µg/µL) as control for non-relevant dsRNA, or iii) a cocktail of dsRNA from *tenecins 1* and *4*, *coleoptericins a* and *b*, and *attacin-2* and then individually transferred to 55mm diameter Petri dishes containing one 15 days-old male, bran flour and gelose for hydration (Fig 6B). Couples were maintained in Petri dishes to allow reproduction until day 7, when females were separated from males and placed in fresh individual Petri dishes containing bran flour and gelose for eggs production. Females were removed from the Petri dishes on day 9 and eggs laid 6 days after immune priming were maintained in Petri dishes for 3–4 days of maturation. Eggs (3–4 days old) were isolated by sieving the flour (mesh size 600 µm) [31] and immediately frozen in liquid nitrogen for future analysis of their antimicrobial activity (Fig 6B). A total of 33 females, leading to the laying of 123 eggs, were used for each treatment (control, GFP-dsRNA and AMPs-dsRNA).

## Temporal dynamics of candidate immune genes expression

To determine the temporal dynamics of the TGIP response in eggs of primed mothers, we assessed the production of the transcripts of candidate immune genes found differentially expressed at the proteomic level in the previous experiments in both females and eggs (Fig 1-4). We especially wanted to quantify the production of these transcripts in both females and eggs at day 1, 5 and 12 post-priming, allowing to have eggs before protection, at maximal protection and when the protection is over, respectively [17] (Fig 6C). In addition, as antibacterial protection in maternally-primed eggs varies with time post-oviposition (eggs age) according to the pathogen that has immune primed the mother [19], we further assessed the effect of eggs age on the amount of transcripts (Fig 6C). For this purpose, groups of 10 days (± 2 days) post-emergence virgin females were, as explained above, immune primed by injection with *B. thuringiensis* (N = 44) or *S. entomophila* (N = 47) or injected with PBS as control (N = 43). Each female was then paired with an immunologically naïve and virgin male of the same age immediately after the priming injection, and allowed to produce eggs. Within each immune treatment modality, a minimum of 9 couples was randomly sacrificed at day 1, 5 and 12 post-priming, while having provided 6 eggs. The remaining couples were transferred into a new Petri dish the day before their sacrifice to control for eggs age. Immediately after the sacrifice of each female, they were stored in liquid nitrogen and from each, 2 eggs were also frozen and

stored in liquid nitrogen. The remaining eggs (2 for each time-point) were allowed to age for 3 days or 7 days post-oviposition before their storage in liquid nitrogen. Hence, within each egg laying sequence, each female contributed to eggs that were allowed to age for 1, 3 or 7 days post-oviposition before to be frozen in liquid nitrogen for later examination. Details of the females, number of eggs used and dates of injection are available in S6 Table.

Total RNA was extracted from both adult females and eggs with Direct-zol RNA MiniPrep kit (Zymo Research, Irvine, CA, USA). Briefly, frozen eggs or adult females were lyzed with a tissue grinder in TRIzol reagent (Life Technologies, Carlsbad, CA, USA). RNA was then purified according to the manufacturer's instruction, including the optional in-column DNAse I treatment, and stored at -80˚C. RNA concentration and purity were controlled by absorbance measurement using an Epoch spectrophotometer (Biotek, Winooski, VT, USA). Reverse transcription of RNA into cDNA was performed with Maxima H minus first strand cDNA synthesis kit (TermoFisher Scientific, Waltham, MA, USA) using random hexamer according to the manufacturer's instructions. Depending on the RNA quantity extracted, 1 µg was used for RT when possible (from adult female an older eggs) or less quantity (from younger eggs).

Quantitative PCR analyses were performed at the "qPCR Haut Débit (qPHD) Montpellier GenomiX" platform using the Labcyte Echo 525 liquid handler for pre-PCR preparation and the LightCycler 480 System (Roche, Basel, Switzerland) for PCR running. PCR reactions were performed in a 1.5 µL total volume comprising 0.5 µL of cDNA (diluted 1:80 or 1:20 with ultra-pure water from adult mother or from egg, respectively) and 0.75 µL of No Rox SYBR Master-Mix dTTP Blue (Takyon Eurogentec, Liege, Belgium), and 100 nM of each primer. PCR amplification efficiencies were established for each target and house-keeping gene by calibration curves using two times serial dilutions of cDNA (from 1:20 to 1:2580) in triplicates. Amplification efficiencies were calculated using slope values of the log-linear portion of the calibration curves by the LightCycler 480 Software release1.5 (Roche). Only primer couples with amplification efficiency of 2 were retained. All details about primers used are reported in S7 Table. The cycling program was as follows: denaturation step at 95˚C for 3 min, 45 cycles of amplification (denaturation step at 95˚C for 10 s, annealing and elongation step at 60˚C for 45s). Quantitative PCR was ended by a melting curve step from 65 to 97˚C with a heating rate of 0.11˚C/s and continuous fluorescence measurement. For each condition, PCR experiments used 6–18 biological replicates of adult females and 3–4 biological replicate of eggs (pool of 6 eggs from 3 different females, 2 eggs per female), in addition to three technical replicates. The mean value of Ct was calculated. Corrected melting curves were checked using the Tm-calling method of the LightCycler 480 Software release1.5. The relative expression of each gene was calculated with the ΔΔCt method as the efficiency of all couple of primers (target and house-keeping genes) presented the same PCR amplification efficiency. For each target gene, the ΔCt was calculated with respect to the mean value of two reference genes coding for 18S and 28S ribosomal RNA. For adult mother, the ΔΔCt were calculated with respect to the ΔCt values of control condition mothers (PBS-injected) sacrificed 1-day post-injection. For the eggs, the ΔΔCt were calculated independently for each laying day, with respect to the ΔCt values of eggs laid by PBS injected mothers. Relative expression values (R) of genes between different conditions were calculated according to the formula $R = 2^{-\Delta\Delta Ct}$ [83].

## Antibacterial assay

Antibacterial activity in the eggs, allowing to check egg protection at the phenotypic level, was measured using a standard diffusion zone assay [16,17,19]. On the one hand, this assay allowed to test the effect of the RNAi invalidation on egg antibacterial activity in the experiment aiming to invalidate the expression of candidate AMPs in eggs laid 5 days post maternal priming and

3 days post-oviposition (Fig 6B). On the other hand, it was used in the experiment allowing to link egg antibacterial activity, from naïve and from PBS, *B. thuringiensis* and *S. entomophila*-treated mothers when laid at 1 and 5 days post priming and 7 days post-oviposition (Fig 6D), with the quantification of AMP candidate gene transcripts in in eggs from similarly treated mothers (Fig 6C).

Individual egg samples were thawed on ice, and egg extracts were prepared by mashing eggs into an acetic acid solution (0.05%, 5 mL per egg). After centrifugation (3,500 g, 2 min, 4˚C), 2 mL of the supernatant was used to measure antibacterial activity on zone of inhibition plates seeded with the bacterium *Arthrobacter globiformis*, obtained from the Pasteur Institute (CIP105365). An overnight culture of the bacterium was added to Broth medium containing 1% agar to achieve a final concentration of $10^5$ cells per mL. Six milliliters of this seeded medium was then poured into a Petri dish and allowed to solidify. Sample wells were made using a Pasteur pipette fitted with a ball pump. Plates were then incubated overnight at 28˚C. The diameter of inhibition zones was then measured for each sample.

## Statistical analyses

Size of egg zones of inhibition following RNA interference treatments were analyzed using Shapiro-Wilk test for normality and Student T-test to test differences in zone of inhibition size ($p < 0.05$), whereas the analysis of the proportion of eggs exhibiting antibacterial activity according to RNA interference treatments were analyzed using Fisher exact tests ($p < 0.05$). Presence of antibacterial activity among the eggs laid by control, PBS-, *B. thuringiensis*- and *S. entomophila*-treated females after 1 day or 5 days post-priming were analyzed using binomial logistic regressions.

Expression of AMPs by RT-qPCR following RNAi treatment in mothers were compared between dsAMP and dsGFP-treated mothers using a Mann Whitney test considering that that data were not normally distributed according to the Shapiro-Wilk test ($p < 0.01$).

The relative expression of the 11 candidate immune genes measured in adult primed females with *B. thuringiensis* and *S. entomophila* and in their 7 days-old eggs laid after 1, 5 and 12 days post-priming relative to PBS-injected control females were analyzed using Mann-Whitney tests ($p < 0.05$), following normality verification using Shapiro-Wilk test. Analyses were made using IBM® SPSS® Statistics 19 software.

## Supporting information

**S1 Fig. Representative examples of variations in AMPs expression levels in primed adult females exposed to dsRNA.** The box plots represent the mean relative expression compared to housekeeping genes (18S ribosomal RNA and Ribosomal protein L27a) for the 5 AMPs tested (tenecin 1 (A), tenecin 4 (B), attacin 2 (C), coleoptericin A (D) and coleoptericin B (E)) following treatment with GFP dsRNA (orange) or with dsAMP (blue). Results of the Mann Whitney test are indicated above box plot for each gene tested, showing no significant reduction of AMP gene expression due to high inter-individual variations following dsRNA injections.
(TIF)

**S1 Table. Selected families of proteins involved in immune and stress responses.** This includes proteases and protease inhibitors, pattern recognition molecules, TOLL pathway, pro-phenoloxidase cascade, immune-responsive effectors, immune regulators, stress and oxidative stress annotated using InterProScan (version 5) with InterPro database version 52.
(XLSX)

**S2 Table. List of manually annotated cytochrome P450.** Clans (mito, 2, 3, or 4), length and sequences are given.
(XLSX)

**S3 Table. List of the proteins identified for each spot differentially abundant in the 2D-DiGE analysis.** This includes the scores of identification of the proteins from each spot and a description of the conserved domain composition of each identified protein.
(XLSX)

**S4 Table. Proteome Discoverer report of the proteins identified in the five egg samples.**
(XLSX)

**S5 Table. Identified peptides in egg samples for the AMP proteins presented in Table 3.**
(XLSX)

**S6 Table. Details of the priming dates, females used and eggs sampled for each experiment.** This includes protein abundance determination by 2D-DiGE, MS/MS identification of AMPs and quantification of candidate immune gene expression by RT-qPCR. For 2D-DiGE, the dye swapping procedure is also indicated. All the combination of females and eggs per female for each biological replicate are indicated. All eggs (laid and in ovaries) for 2D-DiGE and MS/MS analyses were collected 3 days post-priming.
(XLSX)

**S7 Table. List of the primers used for the RT-qPCR experiment.**
(DOCX)

## Acknowledgments

Authors thank Coraline Horin for her participation to the analysis of the transcriptomic database and Dr. David R. Nelson (The University of Tennessee Health Science Center) for the naming of the cytochrome P450 of the transcriptome. They also thank Aurélie Gauthier for critical reading of the manuscript.

## Author Contributions

**Conceptualization:** Guillaume Tetreau, Richard Galinier, David Duval, Philippe Bulet, Christine Coustau, Yannick Moret, Benjamin Gourbal.

**Data curation:** Frédérique Hilliou.

**Formal analysis:** Guillaume Tetreau, Richard Galinier, Yannick Moret.

**Funding acquisition:** David Vaudry, Yannick Moret, Benjamin Gourbal.

**Investigation:** Guillaume Tetreau, Julien Dhinaut, Richard Galinier, Pascaline Audant-Lacour, Sébastien N. Voisin, Karim Arafah, Manon Chogne, Frédérique Hilliou, Anaïs Bordes, Camille Sabarly, Christine Coustau.

**Methodology:** Guillaume Tetreau, Frédérique Hilliou, Philippe Chan, Marie-Laure Walet-Balieu, David Vaudry.

**Project administration:** Guillaume Tetreau, Christine Coustau, Yannick Moret, Benjamin Gourbal.

**Resources:** Philippe Bulet, Christine Coustau, Yannick Moret, Benjamin Gourbal.

**Software:** Philippe Chan, Marie-Laure Walet-Balieu.

**Supervision:** Guillaume Tetreau, Christine Coustau, Yannick Moret, Benjamin Gourbal.

**Validation:** Guillaume Tetreau, Benjamin Gourbal.

**Visualization:** Guillaume Tetreau, Richard Galinier, Frédérique Hilliou, Christine Coustau, Yannick Moret.

**Writing – original draft:** Guillaume Tetreau.

**Writing – review & editing:** Richard Galinier, Sébastien N. Voisin, Karim Arafah, Frédérique Hilliou, Philippe Chan, Marie-Laure Walet-Balieu, David Vaudry, David Duval, Philippe Bulet, Christine Coustau, Yannick Moret, Benjamin Gourbal.

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
