## [Decision Letter · Decision Letter 0]

10 Jun 2020

Dear Dr. TETREAU,

Thank you very much for submitting your manuscript "Deciphering the molecular mechanisms of mother-to-egg immune protection in the mealworm beetle Tenebrio molitor" for consideration at PLOS Pathogens. As with all papers reviewed by the journal, your manuscript was reviewed by members of the editorial board and by several independent reviewers. In light of the reviews (below this email), we would like to invite the resubmission of a significantly-revised version that takes into account the reviewers' comments.

Both reviewers agree that this is an important and interesting topic that is eminently suitable for PLoS Pathogens. However, they have have raised important concerns. In particular, the lack of any immune priming effect in Figure 3B undermines the central conclusion of the manuscript. It seems clear that this could only be addressed using new data. This would also provide the opportunity to address some of their other suggestions about this experiment. For example, using younger eggs before translation of embryonic AMPs could occur would make the interpretation of these results more clear cut and the efficiency of the knock-downs verified. As this likely requires a major experiment to be repeated it is unclear to me whether this work could be completed within the timeframe of a major revision (this could be discussed with the journal staff).

We cannot make any decision about publication until we have seen the revised manuscript and your response to the reviewers' comments. Your revised manuscript is also likely to be sent to reviewers for further evaluation.

Sincerely,

Francis Michael Jiggins

Associate Editor

PLOS Pathogens

Michael Otto

Section Editor

PLOS Pathogens

Kasturi Haldar

Editor-in-Chief

PLOS Pathogens

orcid.org/0000-0001-5065-158X

Michael Malim

Editor-in-Chief

PLOS Pathogens

orcid.org/0000-0002-7699-2064

Both reviewers agree that this is an important and interesting topic that is eminently suitable for PLoS Pathogens. However, they have have raised important concerns. In particular, the lack of any immune priming effect in Figure 3B undermines the central conclusion of the manuscript. It seems clear that this could only be addressed using new data. This would also provide the opportunity to address some of their other suggestions about this experiment. For example, using younger eggs before translation of embryonic AMPs could occur would make the interpretation of these results more clear cut. As this likely requires a major experiment to be repeated it is unclear to me whether this work could be completed within the timeframe of a major revision (this could be discussed with the journal staff).

Reviewer's Responses to Questions

**Part I - Summary**

Reviewer #1: In this manuscript by Tetreau et. al., the authors describe four distinct scenarios which could contribute to trans-generational immune priming (TGIP) against bacterial pathogens in invertebrates: (1) transfer of a “signal” to eggs which could influence gene expression in eggs, (2) transfer of mRNA to eggs, (3) transfer of antibacterial effectors to eggs, and (4) epigenetic modifications. The authors employ a variety of approaches to determine which scenario(s) contribute to TGIP at the egg stage in the eggs of Tenebrio molitor females exposed to Bacillus thuringiensis and/or Serratia entomophila. Based on their data, the authors rule out the possibility that scenarios 1 or 2 substantially contribute to TGIP. In contrast, they suggest that direct transfer of effectors such as antimicrobial peptides does play a role in TGIP (scenario 3). Scenario 4 was not investigated in the present study.

While the authors convincingly demonstrate that scenarios 1 and 2 are unlikely to substantially contribute to TGIP with respect to the 11 genes under consideration, it should be made clear that their results are in fact limited to these 11 genes and should not be interpreted as an indication that scenarios 1 and 2 do not contribute to TGIP as a whole. The results in support of scenario 3 are less compelling. While the results do indeed suggest that AMPs are more abundant in the eggs of primed mothers when the eggs are laid more than three days post-priming (table 2) and that maternally deposited AMPs condition the number of eggs exhibiting antibacterial activity in 3-4 day old eggs when they are laid 4-5 days post-priming (Fig. 3b), it is difficult to link AMP deposition in the eggs, priming of mothers, and TGIP without more appropriate controls. In fact, that the eggs laid by dsGFP-injected, primed mothers were not more likely to exhibit antibacterial activity than the eggs of uninjected, unprimed mothers suggests that TGIP was not even observed in this experiment. The results presented in Fig. 5 are consistent with previous observations of TGIP and support the authors’ conclusions with regard to scenarios 1 and 2, but offer little in the way of mechanistic understanding with regards to scenario 3 given the shortcomings of the experiment depicted in Fig. 3.

Reviewer #2: The manuscript by Tetreau et al. addresses the interesting and important question what the mechanisms underpinning trans-generational immune priming (TGIP) could be. The phenomenon of TGIP describes the enhanced protection of offspring from parents that were immune primed, and has been shown in a number of (mostly) insect species, including the organism that was used here, the mealworm beetle T. molitor. Previous studies that also partly addressed the question of mechanisms of TGIP in general, and in this species in particular, are rather incomplete and give an inconsistent picture. It was not clear whether epigenetic processes are involved, or immune eliciting components (e.g. bacteria-derived substances) are transferred to eggs and/or offspring (leading to stronger activation of egg/offspring immune responses), or whether immune effectors are directly provided into the eggs. The present manuscript uses a large set of experiments and in particular proteomic approaches, showing that the latter possibility is the most relevant for this system. This is the first time, that such a complete data set and combination of experiments was used to address this question. It should be noted, that it is hard to fully exclude the other possibilities as (additional) mediators of TGIP, and in particular for other species. Also, only maternal TGIP can be explained with these mechanisms, not paternal TGIP, a phenomenon also shown for this and other species. However, even in light of these limitations, the strength of the data and the completeness of the study for this system, which is the probably most important model at the moment for maternal TGIP, makes this study very valuable.

**Part II – Major Issues: Key Experiments Required for Acceptance**

Reviewer #1: In addition to the general concerns described above, the authors must address the following major concerns:

Figure 3: The authors should indicate the threshold zone of inhibition required to classify an egg as exhibiting antibacterial activity. For example, all eggs with zone of inhibition > 0 mm or all eggs with zone of inhibition > 1 mm. See also Fig. 5.

Figure 3: The results presented in this figure do not support the hypothesis that priming-influenced maternal deposition of AMPs in the eggs conditions the number of eggs exhibiting antibacterial activity. The eggs of the primed, dsAMP injected mothers were less likely to display antibacterial activity than the eggs of primed, dsGFP injected mothers, suggesting that AMPs deposited by the mother do indeed influence antibacterial activity in the eggs. However, it is difficult to link antibacterial activity in the eggs to priming of the mother because the eggs of primed, dsGFP injected mothers were not more likely to display antibacterial activity than the eggs of unprimed, uninjected mothers. Useful controls for this experiment would be to also use eggs from unprimed, dsAMP injected mothers and from unprimed, dsGFP injected mothers. This would allow one to more directly evaluate the effects of priming. This experiment could also be greatly improved by testing the antibacterial activity in eggs on the same day they are laid. The interpretation of the present results is muddied by the developmental processes occurring in the eggs during the 3-4 day maturation period. If TGIP really is due to maternal deposition of AMPs, then this should be evident in eggs at the moment they are laid. Additionally, the authors should consider the possibility that reducing the expression of AMPs in the mothers may limit the ability of the mothers to respond to the infection and properly promote TGIP in the eggs. Thus, it is possible that the observed reduction in the number of eggs exhibiting antibacterial activity in the dsAMP treatment may only be indirectly related to the levels of AMPs in the mothers. Finally, the authors should present data indicating the efficiency of gene knockdown in the dsAMP vs. dsGFP mothers.

Line 455-457: “Our RT-qPCR results demonstrated that the level of immune gene transcripts in young eggs (1 and 3 days old) was too low to significantly affect the amount of effectors and sustain antibacterial activity in eggs.” This is not demonstrated by the data. The RT-qPCR data demonstrates that the expression of immune genes was below the detection threshold in ~90% of 1 day old eggs, below the detection threshold in ~70% of 3 day old eggs, and near the detection threshold in 1 and 3 day old eggs in which expression could be detected. The expression levels that would be required to “significantly affect the amount of effectors” and how these levels relate to the detection thresholds of the RT-qPCR assays are not discussed.

Lines 478-: “considering that 7-days old eggs from 1-day primed females did not exhibit an increase in antibacterial activity…” There is a problem with the wording here. The authors do not report increases or decreases in antibacterial activity in the eggs. Instead they report the proportion of mothers laying eggs with antibacterial activity.

Lines 539-540: It’s not clear how the deduced protein sequences were obtained. FrameDP is a translation tool that could be used to translate predicted coding sequences in the transcriptome, but Uniprot is a website that provides access to various tools and databases. How exactly was Uniprot used in this process?

Line 672: It is stated here that control females were injected with PBS. This seems to contradict line 1005, which states that control females were not injected.

Lines 717-718: This information is missing from table S6. There is an RT-qPCR page in the excel spreadsheet, but it is blank.

Figure 5: The authors should define what is meant by “protected eggs”. Is this eggs with any zone of bacterial inhibition?

Reviewer #2: It is somewhat surprising that the transcriptomic approach was not used for assessing gene expression differences in primed mothers and eggs. Rather, it seems that the transcriptome that was constructed here was only used to assist protein identification in the proteomic data. It would be straightforward to make use of the basis that was provided here to also explore more deeply the gene expression differences following priming. Instead, gene expression differences where here studied with a rather limited approach, focusing on a few candidate genes (Fig. 4). The results obtained by this approach are interesting and support the conclusions drawn from the proteomic approach, but it would be good to obtain a clearer picture that could e.g. hint why AMPs were down-regulated in eggs at day 5 (which is somewhat surprising). However, given that the proteomic data already provides a lot of information, I probably wouldn't say that this additional experiment is absolutely required, but it is still surprising not find any such data in the manuscript.

The RNAi approach lacks confirmation that it really leads to knock-down of the target genes. A cocktail of 5 dsRNAs were used here, and I am a bit sceptical that all these genes were down-regulated? Using so many genes might also increase the risk of off target effects. More information, in particular RT-qPCR data is needed here.

**Part III – Minor Issues: Editorial and Data Presentation Modifications**

Reviewer #1: I also have some minor concerns the authors may wish to address:

The manuscript is generally clearly written, but throughout the manuscript text and figures the authors should take care to consistently note the age of eggs and the number of days post-priming at which the eggs were laid. The reader is forced to frequently consult the methods section and figure 6 to interpret many of the results and understand the experimental setup. For example, none of this information is provided in Fig. 3 or the figure legend for this figure.

Line 174: I suggest the authors use the ExN50 value, rather than the N50 value, as an assessment of the quality of their de novo transcriptome assembly. The N50 value is more appropriate for genome assemblies and can be a misleading statistic for transcriptome assemblies.

Table 2: With the exception of catalase in the control>bacteria row, no proteins are seen as significantly differentially abundant in both eggs laid and eggs in ovaries. Is this to be expected?

Lines 518-523: The use of insects of various developmental stages and physiological conditions is reasonable, but the authors should be more specific in their description of the insects used to create the sequencing libraries. How many males and females? How many insects from each developmental stage? How many bacterially or phenobarbital challenged individuals were used and at what timepoint post challenge were they collected?

Lines 527 and 532: I suggest the authors specify that these read counts refer to paired-end reads (i.e. the total read count is twice the stated number). This would facilitate comparison to table 1, where total read count is given.

Line 532: The authors state that they sequenced two libraries and after quality control, these libraries contained 58 and 44 million reads (102 million reads). Based on table 1 it seems that these reads were pooled into one set of reads for de novo assembly? If so, this should be explicitly stated.

Line 539: “Traduction” seems to be a French word. Do the authors mean “translation”? See also line 615.

Line 539: “FrameDB” should be “FrameDP”.

Line 555-567: For the 2-D-DiGE experiment, the authors state that laid eggs were collected within 16 hours following oviposition and that collection began after 3 days post-priming. Given that the number of days post-priming at which an egg is laid seems to have important implications for the antibacterial potential of the egg, it would be very useful for the reader if table S6 included a column indicating the number of days post-priming at which each egg was collected. Additionally, it is not clear in the manuscript text or in table S6 at which day(s) post-priming eggs in ovaries were collected. Similar information should be provided for eggs collected for nano-LC-MS/MS (lines 631-635).

Line 1010: The authors should indicate the statistical test used. On line 339 it is stated they a used a Fisher’s exact test, but this should be restated in the figure legend.

Line 686: I suggest the authors refer specifically to the dsRNAs injected rather than “respective knocking down treatments”.

Line 690-693: The description of the egg collection scheme is difficult to understand. As far as I can tell, eggs were laid on days 8 and 9 (corresponding to 4 and 5 days post immune priming, respectively) and then allowed to mature until day 11, at which point the eggs were either 3 or 4 days old. This is all there in the manuscript, but not conveyed in the clearest manner.

Line 730: There is a typo (plateform vs. platform).

Reviewer #2: Summary: It generally captures very well the content of this study, just a detail:

- In my view, the wording is sometimes a bit exaggerated. For example, I would not call this a 'global multidisciplinary approach'.

Introduction:It captures the literature and open questions very well.

Results & Discussion: This section generally reads well, but might be shortened to make it more consistent. I only have a few suggestions:

- Make clear that the intention of producing the transcriptomic data was here only for assisting the proteomic approach. Because of this, the part on the transcriptomic data could also be shortened. I am not sure whether Table 1 needs to be in the main paper, maybe something for the supplement?

- I suggest somewhat more general explanations of the proteomic techniques that were used here, because some readers will not be specialists for these techniques. What is e.g. meant with 'top-down proteomics analysis of eggs extracts' (l. 291-2)?

- It is not always clear how many samples or replicates were used. Please add this important information also to the R&D section (not only the methods part).

- In the last section (l. 423 ff.) it should be made clearer that the literature is sometimes on data obtained in other species. For example, Knorr et al (2005) was on Tribolium castaneum, i.e. even though these are both coleopterans, still other mechanisms might be relevant in this species than in Tenebrio molitor. Other data was even obtained in honey bees, etc. Moreover, the mechanisms relevant here cannot be relevant for paternal TGIP. In summary, it cannot be excluded that different species employ different mechanisms for TGIP, and this should be more clearly stated. This also relates to the Abstract: is T. molitor really a 'model' for clarifying TGIP mechanisms (on a general level), or do we need to be very modest and it is only maternal TGIP to eggs in T. molitor where the mechanism could be clarified here? In my view, this doesn't make the data less valuable.

Materials and Methods:

- l. 497: please provide information about the B.t. strain. Was it a B.t. tenebrionis? Which Cry toxins does it have? It is not straightforward to get this information via the strain number CIP...

- l. 501 and other places: I suggest using the term 'priming' when inactivated pathogens are used, and challenge for life pathogens (which were not used here). It makes more sense given the whole phenomenon is termed TGIP.

- provide more details on the statistical analyses

Figures:

Figure 3: make clear that 3A does not show values with zero (below detection) inhibition

Fig. 5: show statistical analysis

Supplementary files:

Table S6: the tab 'RT-qPCR' seems to be empty

PLOS authors have the option to publish the peer review history of their article (what does this mean?). If published, this will include your full peer review and any attached files.

Reviewer #1: No

Reviewer #2: No
---

## [Decision Letter · Decision Letter 1]

7 Aug 2020

Dear Dr. TETREAU,

Thank you very much for submitting your manuscript "Deciphering the molecular mechanisms of mother-to-egg immune protection in the mealworm beetle Tenebrio molitor" for consideration at PLOS Pathogens. As with all papers reviewed by the journal, your manuscript was reviewed by members of the editorial board and by several independent reviewers. In light of the reviews (below this email), we would like to invite the resubmission of a significantly-revised version that takes into account the reviewers' comments.

The reviewers are satisfied with all but one of the responses. One reviewer felt that the failure to validate the knock-down experiments by qPCR meant no useful conclusions could be drawn from this data and therefore the manuscript should not be published in its current form. I note the other reviewer raised this issue in their initial review and was satisfied with the response. Having read your covering letter, I have sympathy with the frustrations of this experiment. I also agree that the GFP dsRNA control coupled with a rather specific assay for AMP activity provides an indication that RNAi is affecting AMP expression in some way. At a minimum, I would like to see the box plots presented in the covering letter and an accompanying statistical analysis as a supplementary figure. This will make explicit the issues wich are described in the results. Furthermore, in the final Discussion section this issue should be mentioned again, and it should be made clear to what extent the major conclusions depend on this result. However, the ideal validation of knock-down efficiency is to measure protein rather than mRNA levels. Given that AMP concentration can be measured using inhibition zone assays, is there any reason not to check that the treatment reduced AMP concentration in the mothers in this way? This appears to me to be a way to bypass the frustrations of the qPCR, and make the point that if there are fewer of these molecules in the mother there are fewer in the egg.

There are a number of minor comments to address. In addition it seems you may not have seen the comment on data availability on the first draft:

"It seems that the authors haven't seem my previous comment regarding public access to data: For the transcriptome data, I wasn't sure whether the availability through the laboratory website is sufficient? Maybe it should also be put on one of the commonly used repositories?"

I noticed that there was a number looking somewhat like an SRA accession number on the subission form. The sequencing data either needs to be submitted to the SRA, or, if this has been done, it needs to be made clear.

It is PLoS policy to avoid repeated rounds of revisions. I therefore apologise that I have recommended a 'Major Revision'. However, this provides an opportunity to do more experiments  if these are feasible, and  a final decision may not require further peer review. If the experiment I suggested is either impractical or inappropriate, then I can rapidly reach a decision on a revised manuscript.

We cannot make any decision about publication until we have seen the revised manuscript and your response to the reviewers' comments. 

Sincerely,

Francis Michael Jiggins

Associate Editor

PLOS Pathogens

Michael Otto

Section Editor

PLOS Pathogens

Kasturi Haldar

Editor-in-Chief

PLOS Pathogens

orcid.org/0000-0001-5065-158X

Michael Malim

Editor-in-Chief

PLOS Pathogens

orcid.org/0000-0002-7699-2064

The reviewers are satisfied with all but one of the responses. One reviewer felt that the failure to validate the knock-down experiments by qPCR meant no useful conclusions could be drawn from this data. I note the other reviewer raised this issue in their initial review and was satisfied with the response. Having read your covering letter, I have sympathy with the frustrations of this experiment. I also agree that the GFP dsRNA control coupled with a rather specific assay for AMP activity provides an indication that RNAi is affecting AMP expression in some way. At a minimum, I would like to see the box plots presented in the covering letter and an accompanying statistical analysis as a supplementary figure. This will make explicit the issues wich are described in the results. Furthermore, in the final Discussion section this issue should be mentioned again, and it should be made clear to what extent the major conclusions depend on this result. However, the ideal validation of knock-down efficiency is to measure protein rather than mRNA levels. Given that AMP concentration can be measured using inhibition zone assays, is there any reason not to check that the treatment reduced AMP concentration in the mothers in this way? This appears to me to be a way to bypass the frustrations of the qPCR.

There are a number of minor comments to address. In addition it seems you may not have seen the comment on data availability on the first draft:

"It seems that the authors haven't seem my previous comment regarding public access to data: For the transcriptome data, I wasn't sure whether the availability through the laboratory website is sufficient? Maybe it should also be put on one of the commonly used repositories?"

I noticed that there was a number looking somewhat like an SRA accession number on the subission form. The sequencing data either needs to be submitted to the SRA, or, if this has been done, it needs to be made clear.

It is PLoS policy to avoid repeated rounds of revisions. I therefore apologise that I have recommended a 'Major Revision'. However, this provides an opportunity to do more experiments to validate if these are feasible, and it is likely a final decision will not require further peer review.If the experiment I suggested is either impractical or inappropriate, then I can rapidly reach a decision on a revised manuscript.

Reviewer's Responses to Questions

**Part I - Summary**

Reviewer #1: My most significant concern with the original manuscript was with regard to the experiment depicted in Figure 3. Here the authors note that the legend for this figure contained an error which has been corrected in the revised manuscript. As a result, I no longer have concerns regarding the existence of immune priming in the experiment depicted by this figure.

Reviewer #2: With the current revision, the authors have improved the manuscript. They could also clarify several issues in their response letter. No additional experiments were done, but the arguments against doing so are overall convincing. I have a few remaining, minor points regarding the authors' responses and wording in the summary of the manuscript.

**Part II – Major Issues: Key Experiments Required for Acceptance**

Reviewer #1: The authors have appropriately addressed all of my other concerns with the exception of the need to validate their knockdown of AMP transcripts by RNAi. The authors state that they were unable to validate the knockdown due to high inter-individual variation in AMP expression. From the figure included in the response to reviewers, it is indeed clear that there is very high variation in the expression of AMPs between individuals. Based on this figure, it seems that many dsGFP- and dsAMP-injected individuals have highly reduced AMP expression ratios and there is no discernable difference in AMP expression between dsGFP- and dsAMP-injected adults. The authors state that “even in the worst-case scenario where the silencing of AMPs is limited, we would only underestimate the effect of AMPs on the TGIP phenotype..”. Indeed a change to the TGIP phenotype is observed in the dsAMP-injected mothers compared to the dsGFP-injected mothers, but without proper validation of the AMP knockdowns, it is not appropriate to link this phenotype to reduced expression of AMPs. In the context of such high variation in AMP expression and the use of a relatively small number of mothers for this experiment, the need to validate the knockdown is particularly important.

Reviewer #2: N/A

**Part III – Minor Issues: Editorial and Data Presentation Modifications**

Reviewer #1: (No Response)

Reviewer #2: Summary:

Only minor changes were made to the summary upon my criticism that the wording in the summary is in some places a bit exaggerated. However, I am afraid that I am not fully happy with these changes either. The word 'comparative' might indicate to some readers that this would be a study based on comparative approaches (i.e. species comparisons), which is clearly not the case here.

Moreover, the sentence 'present data clearly reject the involvement of mRNA' is, in my view, too strong, given the limitations of the approach that could only address mRNA for some candidate genes (I suggest deleting 'clearly').

Authors' general response #(1):

While the clarification helps understanding the data in Fig 3 (and conclusions drawn from it), it should be noted that a comparison of primed with unprimed mothers is missing; ie. the full design would include priming and RNAi as factors. I suggest to add this consideration to the paper.

Authors' general response #(2):

I appreciate the detailed discussion of the issues with validation of the RNAi using a cocktail of dsRNAs. I still consider these methodological problems a little worrying, though. The huge variability in the qPCR data used in the attempt to validate the RNAi effect could be a methodological problem (low repeatability of the assay) or large inter-individual variation (e.g. because the beetle population is diverse). Any chance to discriminate between these possibilities (e.g. repeatability of sample replicates)?

Material and Methods, my question regarding the Bt strain used:

I am not convinced by the argument that Bt strain information was not provided because 'it is not a relevant information in the context of this article'. The methods of a paper should be as precise as possible, e.g. to enable reproduction of the study. There are many different strains of Bt, e.g Bt tenebrionis which is infective to beetles, while other strains are not (in oral infection). Priming can be specific regarding the Bt strain in both oral and septic infections, thus this information is not irrelevant.

PLOS authors have the option to publish the peer review history of their article (what does this mean?). If published, this will include your full peer review and any attached files.

Reviewer #1: No

Reviewer #2: **Yes: **Joachim Kurtz
---

## [Editor Report · Decision Letter 2]

28 Aug 2020

Dear Dr. TETREAU,

We are pleased to inform you that your manuscript 'Deciphering the molecular mechanisms of mother-to-egg immune protection in the mealworm beetle Tenebrio molitor' has been provisionally accepted for publication in PLOS Pathogens.

Best regards,

Francis Michael Jiggins

Associate Editor

PLOS Pathogens

Michael Otto

Section Editor

PLOS Pathogens

Kasturi Haldar

Editor-in-Chief

PLOS Pathogens

orcid.org/0000-0001-5065-158X

Michael Malim

Editor-in-Chief

PLOS Pathogens

orcid.org/0000-0002-7699-2064

It is unfortunate that it was not possible to validate the RNAi efficiency, but I can empathise with the logistical constraints. However, as the text is now very cautious about these results and the phenotype measured is very specific to AMPs I do not feel this is a fatal flaw. Together the results are an important contributon to our understanding of immune priming.
---

## [Editor Report · Acceptance letter]

8 Oct 2020

Dear Dr. TETREAU,

We are delighted to inform you that your manuscript, "Deciphering the molecular mechanisms of mother-to-egg immune protection in the mealworm beetle *Tenebrio molitor*," has been formally accepted for publication in PLOS Pathogens.

Best regards,

Kasturi Haldar

Editor-in-Chief

PLOS Pathogens

orcid.org/0000-0001-5065-158X

Michael Malim

Editor-in-Chief

PLOS Pathogens

orcid.org/0000-0002-7699-2064